



# New water fractions and transit time distributions at Plynlimon, Wales, estimated from stable water isotopes in precipitation and streamflow

Julia L. A. Knapp[1], Colin Neal[2], Alessandro Schlumpf[3], Margaret Neal[2], and James W. Kirchner[1,3,4]

[1]Department of Environmental Systems Science, ETH Zurich, 8092 Zurich, Switzerland
[2]Centre for Ecology and Hydrology, Wallingford, OX10 8BB, UK
[3]Swiss Federal Research Institute WSL, 8903 Birmensdorf, Switzerland
[4]Department of Earth and Planetary Science, University of California, Berkeley, CA 94720, USA

*Correspondence to*: Julia L. A. Knapp (julia.knapp@usys.ethz.ch)

**Abstract.**

Long-term, high-frequency time series of passive tracers in precipitation and streamflow are essential for quantifying catchment transport and storage processes, but few such data sets are publicly available. Here we describe, present, and make available to the public two extensive data sets of stable water isotopes in streamflow and precipitation at the Plynlimon

experimental catchments in mid-Wales. Stable isotope data are available at 7-hourly intervals for 17 months, and at weekly intervals for 4.25 years. Precipitation isotope values were highly variable in both data sets, and the high temporal resolution of the 7-hourly streamwater samples revealed rich isotopic dynamics that were not captured by the weekly sampling.

We used ensemble hydrograph separation to calculate new water fractions and transit time distributions from both data sets.

Transit time distributions estimated by ensemble hydrograph separation were broadly consistent with those estimated by spectral fitting methods, suggesting that they can reliably quantify the contributions of recent precipitation to streamflow. We found that on average, roughly 3% of streamwater was made up of precipitation that fell within the previous 7 hours, and 13-15% of streamwater was made up of precipitation that fell within the previous week. The contributions of recent precipitation to streamflow were highest during large events, as illustrated by comparing new water fractions for different

discharges and precipitation rates. This dependence of new water fractions on water fluxes was also reflected in their seasonal variations, with lower new water fractions and more damped catchment transit time distributions in spring and summer compared to fall and winter.

We also compared new water fractions obtained from stable water isotopes against those obtained from concentrations of

chloride, a solute frequently used as a passive tracer of catchment transport processes. After filtering the chloride data for dry deposition effects, we found broadly similar new water fractions using chloride and stable water isotopes, indicating that these



different tracers may yield similar inferences about catchment storage and transport, if potentially confounding factors are eliminated.

These stable isotope time series comprise some of the longest and most detailed publicly available catchment isotope data sets.
They complement extensive solute data sets that are already publicly available for Plynlimon, enabling a wide range of future analyses of catchment behavior.

## 1 Introduction

Passive tracers have frequently been used to understand transport and mixing processes at the catchment scale. Because these tracers do not react with their environment, but instead are transported with the water, their time series in streamflow and
precipitation can be compared to estimate transit time distributions and timescales of catchment storage (Christophersen and Neal, 1990; Hrachowitz et al., 2009). Due to the cost and effort involved in collecting and analyzing environmental tracers, available data sets typically comprise either short series of high-frequency measurements, or longer series of lower-frequency measurements. High-frequency tracer time series measured over a few days to weeks can help in understanding storage, mixing and transport processes during individual storm events, and how these change with factors such as precipitation intensity and
catchment wetness (e.g., Casper et al., 2003; James and Roulet, 2009; Segura et al., 2012). Conversely, tracer measurements covering several seasons or years at weekly or bi-weekly resolution can help in understanding inter-seasonal changes in storage, long-term effects of disturbance events, and implications of climate and land-use change (e.g., McGuire et al., 2002; Heidbüchel et al., 2013).

Few catchment tracer time series are publicly available. This is particularly true for time series of stable water isotopes, which are nearly ideal passive tracers because they are part of the water molecule itself. Among the few long-term time series that are publicly available, for example, are isotope measurements taken biweekly from 2006 to 2010 at Hubbard Brook Watershed 3 (Campbell and Green, 2019). Stable water isotope time series are also publicly available for many sites worldwide through the Global Network of Isotopes in Precipitation (GNIP) and Global Network of Isotopes in Rivers (GNIR), both hosted by the
International Atomic Energy Agency (IAEA, 2019; IAEA/WMO, 2019). These time series, however, mostly consist of only a few samples, and often have large gaps.

If we want to exploit the full potential of stable isotopes to trace flowpaths and quantify travel times, we need to sample them at much higher than weekly frequencies. This requires longer-term, higher-frequency records than are currently available.
Because similar records of stable water isotopes are scarce, many analyses of catchment transport, storage and mixing have used anion tracers like chloride instead (e.g., Duffy and Gelhar, 1986; Kirchner et al., 2000; Hrachowitz et al., 2009; Remondi et al., 2018). This raises the obvious question of whether analyses of chloride and isotope tracers yield comparable results





(Neal and Rosier, 1990; Kirchner et al., 2010), as neither tracer can be considered a gold standard. Stable water isotopes suffer from evaporative fractionation, which may be inconsequential if the evaporated waters then evaporate completely, but may substantially affect analyses of catchment processes if the fractionated water constitutes a non-trivial fraction of stream discharge. Stable water isotope data also need to be corrected for altitude effects if the precipitation sampling point is not

representative of the average catchment elevation (Dansgaard, 1961; Clark and Fritz, 2013). Unlike stable water isotopes, chloride concentrations can be affected by dry deposition between rain events. These dry deposition inputs and their effect on the catchment input-output relationship are difficult to measure and quantify (Juang and Johnson, 1967; Durand et al., 1994; Guan et al., 2010). Furthermore, chloride may be affected by evapoconcentration, undergo ion exchange buffering, and interact biogeochemically with vegetation and soils (Öberg, 2002; Lovett et al., 2005; Bastviken et al., 2007). In some catchments,

anthropogenic contamination with chloride from road salt or fertilizers also may be substantial. Thus chloride is most useful as a tracer where sea salt inputs are high enough, and variable enough, to overwhelm these potentially confounding factors. Stable water isotopes, on the other hand, will be most useful where the isotopic input signals are highly variable over time, due to catchment inputs alternating among isotopically distinct source regions and atmospheric pathways with varying degrees of Rayleigh distillation.

Because they are shaped by different processes, atmospheric inputs of chloride and water isotopes are only weakly correlated and follow very different distributions. Thus direct comparisons of chloride and water isotope time series are not informative. However, both tracers should yield similar inferences about catchment storage, transport, and mixing, if they are both transported conservatively with the water.

In this manuscript we document, present, and make available to the public two extensive data sets of stable water isotope time series recorded in precipitation and streamwater at the intensively studied Plynlimon experimental catchments in Wales; one at weekly resolution for 4.25 years, and the other one at 7-hourly intervals for 17 months. We present details of these data sets and their collection and analysis. For both data sets, associated solute data sets are already available, spanning a range of

solutes including chloride.

We furthermore use these isotope time series to quantify the relative amount of streamflow that is made up of recent precipitation, by applying the recently developed "ensemble hydrograph separation" approach of Kirchner (2019). This approach assesses catchment transport and mixing processes by quantifying, directly from measured data, the fraction of

streamwater that fell as precipitation during the last sampling interval. This "new water fraction" has previously been validated through benchmark testing with synthetic isotope time series, but has not yet been applied to real-world tracer data. We additionally investigate the sensitivity of new water fractions to discharge, precipitation intensity, and time of year, and determine transit time distributions using both ensemble hydrograph separation and power spectrum fitting. We compare results from weekly and 7-hourly sampling, to better understand mixing and storage processes in this extensively studied

catchment. We also compare these results to those obtained from chloride time series, to determine whether both passive tracers yield similar inferences about the storage and release of water from the catchment. Through applying ensemble hydrograph separation to these unique tracer data sets, we illustrate their usefulness for understanding storage and mixing processes at Plynlimon.

**2 Site description**

The Plynlimon catchments comprise the uppermost headwaters of the river Severn in mid-Wales, and are situated approximately 20 km inland from the coast. Since the 1960's, Plynlimon has been a focal point for studying the effects of plantation forestry and climate change on the water cycle (Kirby et al., 1991; Durand et al., 1994; Neal et al., 2001; Neal et al., 2003). As part of this research, precipitation and streamflow have been sampled at Plynlimon since the 1980's and analyzed
for an unusually wide range of solutes; the resulting publicly available chemical time series are unique worldwide (Neal et al., 2013b, c; Norris et al., 2017).

Plynlimon comprises several catchments that differ in land use and elevation, covering a combined area of 19.25 km$^2$ and ranging in elevation from 319 to 738 m a.s.l. In this study, we present measurements of stable isotopes and chloride in
precipitation as well as in streamflow from three of these catchments, Upper Hafren, Lower Hafren, and Tanllwyth (1.22 km$^2$, 3.58 km$^2$, and 0.92 km$^2$, respectively).

The bedrock at Plynlimon is composed of Lower Paleozoic rocks overlain by acidic soils typically less than 1 m thick. Paleozoic grits, mudstones and shales form the parent material of these soils, and soil differentiation depends on drainage.
While podzolized soils dominate freely draining areas, blanket peats are found in areas with impeded drainage at higher altitudes (Kirby et al., 1991).

The Upper Hafren catchment consists mainly of semi-natural moorland used for sheep grazing. The Lower Hafren and Tanllwyth catchments, by contrast, are covered by coniferous plantation forests (mainly Sitka spruce), which were planted in
the 1940s-1960s and have been subjected to phased felling and clearfelling with subsequent replanting over the years (Neal et al., 2001; Neal et al., 2003; Neal et al., 2004a, b). In establishing the forest plantations, soils were plowed and networks of ditches were dug to increase drainage and minimize waterlogging (Neal et al., 2004b). Although large areas of these catchments have since been felled and replanted, these drainage ditches remain, creating flashy hydrograph responses to rainfall (Kirby et al., 1991; Leeks and Marks, 1997; Marks and Rutt, 1997). The climate at Plynlimon is humid and cool, with annual
precipitation of approximately 2400 mm, and monthly mean temperatures around 2-3 °C in winter and 11-13°C in summer (Kirby et al., 1991).





## 3 Description of the data set

### 3.1 Sample collection and analysis

A total of 607 precipitation and streamwater samples were collected at weekly intervals between December 2004 and March 2009. Precipitation was sampled as cumulative bulk samples at Carreg Wen, and streamwater was collected as instantaneous

grab samples at the Lower Hafren and Tanllwyth sampling points (see Fig. 1). The precipitation samples were collected with a continuously open PVC funnel of 15 cm diameter with anti-bird protection. This data set is termed the "weekly data" in our analysis.

A further 2113 samples were collected at 7-hourly intervals from July 2007 through March 2009 at Carreg Wen (precipitation)

and the Upper Hafren outlet (streamwater). Streamwater sampling was automated using Xian 1000 portable automatic samplers programmed to collect streamwater at 7-hourly intervals into carousels of 24 500 ml bottles, which were picked up from the field site once per week (for details see Neal et al., 2012; Neal et al., 2013a). The entire sampling pathway was flushed with stream water immediately before the collection of each sample to avoid carryover from the previous sample. The precipitation samples were collected via a continuously open 57.5-cm funnel with anti-bird protection, mounted above an autosampler with

an enclosed carousel of 24 308-ml bottles, which was also picked up once per week. The resulting data set is termed the "7-hourly data" in our analysis.

Both the weekly and 7-hourly sampling of stable water isotopes were embedded in longer-term data collection efforts which have previously been published. Stream chemistry analyses are available weekly from 1983 through 2011 at up to five

catchments (Neal et al., 2011; Neal et al., 2013b), and bi-weekly thereafter (Norris et al., 2017), and are also available at 7-hourly resolution between 2007 and 2009 (Neal et al., 2012; Neal et al., 2013c; Neal et al., 2013a). In addition, hourly meteorological measurements and 15-minute stream gauging data from Plynlimon are available starting in the 1970's (CEH, 2019). Thus the isotope measurements presented in this study complement chemical and hydrological data already available to the public.

Rainfall amounts were recorded from a standard, ground-level tipping bucket rain gauge at the Tanllwyth met site during the weekly sampling (at 350 m a.s.l.), and at the Carreg Wen automatic weather station during the 7-hourly sampling (at 575 m a.s.l.). Streamflow was determined from stream gauges at the outlets of the Upper Hafren, Hafren, and Tanllwyth catchments.

Each sample bottle (weekly sampling) or carousel of bottles (7-hourly sampling) was normally processed at the Center for Ecology and Hydrology (CEH), Bangor, UK the day after it was returned from the field. All samples were filtered (0.45 μm Supor membrane) and then split. One part of the sample was acidified to 1% v/v with concentrated high-purity HNO₃ for analysis of cations and metals, and another was bottled without acidification for analysis of anions. A 30-ml aliquot was also



bottled in high-density polyethylene bottles for subsequent isotopic analysis. Usually this aliquot was taken from the un-acidified split, but in some cases it was taken from the acidified split by mistake. The behavior of the acidified and un-acidified samples was broadly similar (see the supplement, including supplemental figure S2, for more information), but the acidified samples were slightly lighter in deuterium than the un-acidified samples. This offset has been corrected in the data set, and the

acidified samples are flagged in the archival data file provided as supplemental material to this paper.

The bottled isotope samples were kept in the dark at or below 5°C until they were shipped from CEH to the central laboratory of the Swiss Federal Institute for Forest, Snow and Landscape Research (WSL), Birmensdorf, Switzerland, in 2009. At WSL, all samples were transferred to 2 ml glass vials and closed with 11 mm snap caps (Infochroma AG, Goldau, Switzerland) with

1 mm silicone septa.  The vials were stored at -14°C for about 1 year, before they were thawed for subsequent analysis.

The samples were analyzed for oxygen-18 and deuterium isotope ratios at WSL, using a Picarro L1102-i cavity ring-down spectroscopy (CRDS) analyzer equipped with a Picarro V1102-i vaporization module (Picarro, Inc., Sunnyvale, CA, USA) and a PAL HTC-xt-LEAP-Pic autosampler (CTC Analytics AG, Zwingen, Switzerland). Routine calibrations used three

secondary standards (mixed seawater, Fiji artesian water, and Sion drinking water), which in turn were referenced to IAEA VSMOW2, SLAP2, and GISP. All isotope ratios are reported in standard δ-notation relative to the Vienna Standard Mean Ocean Water (VSMOW). Each sample was analyzed twice, separated by at least 100 other samples and standards, and the reported value for each sample is the average of the pair. Any pair that differed by more than 0.20‰ in oxygen-18 or 1.0‰ in deuterium was re-analyzed (again twice), and whichever of the two pairs was more consistent was averaged for the final

reported value. Furthermore, the isotopic composition of some streamwater samples was cross-checked using isotope ratio mass spectrometry (IRMS) and a generally good agreement was found to the CRDS measurements (supplemental figure S1). More details on the comparison between IRMS and CRDS measurements, and the drift and reproducibility of the measurements, is given in the supplement.

**3.2 Data set validation and proviso**

The local meteoric water line determined from isotope ratios of the 7-hourly precipitation samples (LMWL, $\delta^2H = 11.8 + 7.71$ $\delta^{18}O$) fell close to the global meteoric water line (GMWL: $\delta^2H = 10 + 8 \delta^{18}O$). We excluded one 7-hourly streamwater sample from further analysis, since its isotope ratio deviated significantly from the GMWL, suggesting evaporation during storage (see supplemental figure S3). In addition, some of the weekly samples showed clear evaporation trends and had to be omitted

from the data set; they were isotopically heavy in both deuterium and oxygen-18, but followed a line that was much shallower than the meteoric water line, indicating evaporative fractionation. In total around one-eighth of the weekly samples had to be excluded from the data set (22/177 precipitation samples, 25/215 Lower Hafren streamwater samples, and 27/215 Tanllwyth streamwater samples). Most of these sample bottles had visually obvious head space when they were opened for analysis,





despite having been completely filled at the time of original sample processing. The dual-isotope plots in supplemental figure S3 show clear evidence of evaporative fractionation in the excluded samples.

Some of the 7-hourly samples were lost due to sporadic autosampler failures, and all samples collected between December 5 2007 and the middle of March 2008 were lost following chemical analysis, resulting in a data gap in the 7-hourly isotope time series. As a result, over half of the 7-hourly streamwater samples are missing during the months of December through March. However, the streamwater samples are 98.6% complete from 18 March through 29 November 2008 (864 samples out of 876 sampling periods) following all quality control checks.

During some sampling intervals, too little rain fell to provide sufficient sample volume, and thus precipitation isotope analyses 10 are missing for some low-volume rainfall events. Conversely, if the rainfall during a seven-hour sampling period exceeded the capacity of the sample bottle (308 ml, which equals 1.2 mm of rain), the bottle overflowed. For such precipitation samples, the isotopic ratios of the sample may differ, by an unknown amount, from the volume-weighted averages over the 7-hour interval. Such overflows occurred during approximately 65% of the 7-hour intervals for which rainfall samples are available. Because 15 these samples also comprise the great majority of the total rainfall, and because within-event variations in precipitation isotopes can be large (Munksgaard et al., 2012; von Freyberg et al., 2017), isotopic mass balances derived from these data should be treated with caution.

The dual-isotope plots in Fig. 2 show that the streamwater and precipitation samples of the final data set fell close to the 20 GMWL, suggesting that they were not greatly affected by evaporative fractionation. Nonetheless, one must consider the possibility that some evaporative fractionation has taken place, particularly in samples collected during the warmer seasons, given that they were stored in the field for up to a week in open bottles within the autosampler. To test for this possibility, we plotted the deuterium excess (Dansgaard, 1964) as functions of season (Fig. 3) and function of the length of time each sample was stored in the field (Fig. 4). More negative values of deuterium excess indicate greater degrees of evaporative fractionation. 25 Figure 3 shows that the deuterium excess in both the 7-hourly and weekly samples was close to 10 (the GMWL constant, indicated by the reference line in Fig. 3), and was only slightly lower in the summer than the winter. Importantly, the seasonal pattern in deuterium excess in the 7-hourly streamwater samples (which were stored for up to a week in the autosampler in the field) was similar to that in the weekly streamwater samples (which were collected by manual grab sampling and brought directly back to the lab). The similarity in these two deuterium-excess patterns implies that the 7-hourly samples did not 30 undergo significant evaporative fractionation while in storage in the autosampler. This inference is corroborated by Fig. 4, which shows that the storage duration in the field had no detectable effect on the deuterium excess in 7-hourly precipitation and streamwater samples, in either the summer or winter seasons.



## 3.3 Characteristics of the data set

Figure 5 shows the 7-hourly and weekly time series of deuterium and oxygen-18 in precipitation and streamwater. The left and right axes are scaled such that fluctuations following the meteoric water line will appear equal, facilitating easier visual comparison. Both isotopes, at both sampling frequencies, show that streamwater isotope variations are very strongly damped
compared to precipitation. This directly implies that recent rainfall can only be a minor component of streamflow. Consequently, streamflow must be composed of a mixture of many previous precipitation inputs, and thus the catchment must store and mix waters over a wide range of time scales.

Neither the weekly nor the 7-hourly data exhibit strong seasonal patterns, reflecting the proximity of Plynlimon to the Irish
Sea. The weekly precipitation isotope measurements are distinctly less variable than the 7-hourly measurements are, because the weekly samples average over higher-frequency isotopic fluctuations that are captured in the 7-hourly samples. The weekly streamflow time series also appears smoother than the 7-hourly time series, but this is largely a visual artifact resulting from the lower density of data points (roughly 200 weekly samples, versus nearly 1500 7-hourly samples, in plots of equal width in Fig. 5). Nonetheless, the 7-hourly streamwater sampling does capture several large brief isotopic excursions that are missed
by the weekly sampling.

Figure 6 presents a close-up of part of the 7-hourly streamwater isotope time series, revealing rich dynamics in streamwater isotopes that are nearly invisible at the scale of the whole record shown in Fig. 5. The fluctuations in deuterium and oxygen-18 generally mirror one another, with distinctly larger excursions during high-flow conditions, providing a first indication that
higher flows contain larger proportions of recent precipitation, and smaller proportions of older catchment storage. The streamwater isotope fluctuations are much smaller at low flows, but they are not noise. This can be seen by comparing the streamwater isotope time series in Fig. 6 to the green lines, which show a reproducibility test in which a single sample was analyzed 124 times in sequence, revealing the variability that would be expected to arise from analytical noise alone. The variability in the streamwater time series is distinctly larger than this, indicating that it mostly reflects real-world variability in
the streamwater isotopes.

The somewhat narrower band of deuterium data, as seen in Fig. 6, presumably reflects the larger measurement noise associated with the oxygen-18 values, or, conversely, the stronger memory effect that arises in deuterium analyses due to heterogeneous exchange of hydrogen with adsorbed water in the analyzer (Friedman and Irsa, 1952). Both of these hypotheses are consistent
with the reproducibility test, in which the replicate oxygen-18 and deuterium measurements had standard deviations of 0.069‰ and 0.22‰, respectively. Thus, in this test, deuterium was only about three times as noisy as oxygen-18, whereas its real-world variability should be eight times larger than that of oxygen-18 for samples that follow the meteoric water line. Thus the signal-





to-noise ratio in deuterium should be roughly twice as large as in oxygen-18, although deuterium's sample-to-sample memory effects were also three times larger (see supplement).

The fluctuation damping in the streamwater isotopes, relative to the much larger fluctuations in precipitation isotopes, can also be visualized through power spectra. As Fig. 7 shows, fluctuations in streamwater isotopes are strongly damped relative to precipitation on all time scales shorter than several years, and the degree of damping systematically grows as frequency increases. There is distinct power-law scaling in both the precipitation and streamwater time series on time scales shorter than roughly one month (corresponding to frequencies higher than roughly 10 per year), with steeper scaling in streamwater than precipitation. The spectral slopes of the two isotopes in 7-hourly precipitation are indistinguishable within error (0.63±0.03 and 0.62±0.02 for deuterium and oxygen-18, respectively).  By contrast, the spectral slope of deuterium in 7-hourly streamwater is distinctly steeper than that of oxygen-18 (1.57±0.07 versus 1.17±0.02), possibly reflecting greater memory effects during the analysis. The spectral slope of deuterium in weekly streamwater is also slightly steeper than that of oxygen-18 at both Lower Hafren and Tanllwyth, but not by more than the standard error.

## 4 Calculation methods

### 4.1 New water fractions and transit time distributions

The new water fraction $F_{\text{new}}$ uses passive tracers to quantify the average contribution of recent precipitation to streamflow across an ensemble of time steps, using the slope of the simple linear regression (Kirchner, 2019):

$$C_{Q_j} - C_{Q_{j-1}} = F_{\text{new}} \left( C_{P_j} - C_{Q_{j-1}} \right) + \alpha + \varepsilon_j \tag{1}$$

where $C_{P_j}$ and $C_{Q_j}$ represent the tracer concentrations (or isotope values) in precipitation and streamwater, respectively, for a series of sampling times $j$; $\alpha$ is the regression intercept, and $\varepsilon_j$ is the error term.  The uncertainty in $F_{\text{new}}$ can be estimated as the standard error of the regression slope of Eq. (1). This so-called "ensemble hydrograph separation" approach is based on the principle that the larger the fraction of recent precipitation in streamflow, the more tightly correlated their tracer concentrations will be.

New water fractions can be calculated to represent different aspects of catchment behavior (Kirchner, 2019). *Event new water fractions* $\left( {}^{Q_p}F_{\text{new}} \right)$ quantify the proportion of new water found in streamflow for time steps with precipitation, whereas *new water fractions for all time steps* $\left( {}^{Q}F_{\text{new}} \right)$ scale the event new water fraction by the proportion of days with precipitation to obtain an average value of new water in streamflow, including rainless periods. Similar to these new water fractions of discharge, the *new water fraction of precipitation* $\left( {}^{P}F_{\text{new}} \right)$ quantifies the fraction of precipitation that becomes streamflow within the given sampling interval (which will generally differ from the fraction of streamflow that is composed of recent





precipitation). These new water fractions can be also weighted by volume, giving more weight to sampling times with higher flow, rather than weighting each time interval uniformly; volume-weighted quantities are indicated by an asterisk, i.e., $^{Qp}F_{new}^*$, $^{Q}F_{new}^*$, and $^{P}F_{new}^*$.

Here we calculate new water fractions from both deuterium and oxygen-18 to gain insights into the responses of the Plynlimon catchments to precipitation. We furthermore compare new water fractions for different seasons and discharge regimes to explore how the catchments' behaviors vary under different conditions. To quantify how much precipitation contributes to streamflow over a range of lag times we also determine "backward" and "forward" transit time distributions (which quantify the relative amount of streamflow that originated as rainfall at different prior times, and the relative amount of precipitation

that will become stream flow at different future times, respectively), by extending Eq. (1) to a multiple regression that accounts for multiple time lags. For documentation of this method, as well as further details on the ensemble hydrograph separation approach, the conditions under which it holds, and the different types of new water fractions, please see Kirchner (2019).

### 4.2 Estimation of transit time distributions from spectra

The spectral damping shown in Fig. 7 can be used to estimate equivalent transit time distributions to those estimated by ensemble hydrograph separation. Our approach is based on the convolution theorem of linear systems analysis, which implies that the power spectrum of the streamwater isotope time series should equal the power spectrum of the precipitation isotope time series, multiplied by the power spectrum of the transit time distribution. We assume that the transit time distribution is approximated by the gamma distribution,

$$p(\tau) = \frac{\tau^{k-1} e^{-\tau/\theta}}{\theta^k \Gamma(k)} \quad , \tag{2}$$

where $\tau$ is the transit time, $\theta$ and $k$ are scale and shape parameters, and $\Gamma(k)$ is the gamma function. The mean transit time can be calculated as $\bar{\tau} = k\theta$, but it will be very sensitive to the upper tail of the distribution, and thus difficult to constrain from relatively short, high-frequency tracer time series. To estimate the parameters $\theta$ and $k$, we multiplied the power spectrum $S_P(\omega)$ of the precipitation isotope time series by the power spectrum of the gamma distribution, to yield an estimate of the

power spectrum $S_Q(\omega)$ of the streamwater isotope time series:

$$S_Q(\omega) = S_P(\omega) [1 + (\omega \theta)^2]^{-k} \quad , \tag{3}$$

where $\omega = 2\pi f$ is the angular frequency. We estimated the parameters $\theta$ and $k$ in Eq. (3) by minimizing the sum of squared deviations between the logarithm of the predicted spectrum $S_Q(\omega)$ and the logarithm of the measured tracer spectrum in streamwater, using the analytic Gauss-Newton algorithm as implemented in JMP v. 14.3 (SAS Institute, Cary, NC, USA).

Weighted transit time distributions can also be estimated from Eq. (3), using a precipitation-weighted spectrum of the precipitation tracer time series, and a discharge-weighted spectrum of the streamwater tracer time series.



### 4.3 Comparison to chloride data

Many studies of catchment transit times and rainfall-runoff processes are based on stable water isotopes. Until recently, however, stable isotope measurements were expensive and therefore relatively rare. Instead, chloride has been widely used as a passive tracer, under the assumption that it is transported conservatively through the catchment. However, dry deposition of aerosols can account for a substantial share of the total chloride input, particularly at catchments like Plynlimon that are close to the coastline (Neal and Rosier, 1990; Neal and Kirchner, 2000). At Plynlimon, dry deposition accounts for about 10-20% of total chloride inputs from the atmosphere (Durand et al., 1994; Wilkinson et al., 1997). Dry deposition greatly increases the variability of the precipitation concentrations, but it probably has a small effect on calculated fluxes, because samples dominated by dry deposition are usually associated with small volumes of water. However, the ensemble hydrograph separation approach uses concentration data, and specifically the damping observed between catchment input and output concentrations, to determine fractions of recent precipitation in streamflow. It is thus highly sensitive to the variability of the input signal. Noise introduced into the input signal through dry deposition can bias the results towards smaller new water fractions, because the higher variability in the input signal incorrectly implies a stronger damping between the input and output signal. To minimize this bias, we excluded all 7-hourly precipitation chloride samples that were potentially influenced by dry deposition according to several soft and hard criteria, e.g., very high concentrations in small sample volumes, or samples immediately following extended intervals without precipitation. The details of this filtering procedure are explained in the supplements.

We did not apply a similar filtering procedure to the weekly precipitation samples, because we expected them to be much less affected by dry deposition. The 7-hourly samples were particularly vulnerable to dry deposition because they were collected using a large funnel, which required only a very small input of liquid precipitation to make a measurable sample volume (e.g., only 0.4 mm of precipitation yielded 100 ml of sample). By contrast, the funnel used for the weekly sampling was substantially smaller, so more wet deposition was required to make a measurable sample, thus providing greater dilution of any dry deposition. Nonetheless, we removed chloride samples with concentrations more than three standard deviations above or below the mean, as potential outliers, from the weekly precipitation and streamwater time series (as well as from the 7-hourly streamwater time series).

### 4.4 Aggregation of sampling intervals

To investigate the scaling of new water fractions with the length of the sampling interval, and to allow a comparison between the three catchments, we combined sequential sets of 7-hourly samples to synthesize longer sampling intervals. In case of the instantaneous streamwater samples, only the grab sample collected at the end of the aggregated sampling interval was considered, and all other samples in between were disregarded. Precipitation samples, on the other hand, are cumulative





samples. Therefore, all individual samples collected during an aggregated sampling interval were averaged together, weighted by their respective precipitation rates.

The longer the aggregated sampling interval, the greater the number of possible combinations of samples. To exclude the
possibility that an arbitrary choice of sample combinations would affect the results, we calculated new water fractions with all possible sample combinations, and averaged the resulting quantities.

## 5 Results and discussion

### 5.1 New water fractions

We calculated new water fractions using deuterium, oxygen-18, and chloride collected at 7-hourly and weekly intervals (Table
1). New water fractions calculated from the 7-hourly isotope data show that slightly less than 3% of streamflow was made up of precipitation that fell within the last 7 h. New water fractions calculated from the weekly isotope data show that roughly 13-15% of streamflow consisted of precipitation that fell within the last week (in both cases, these are volume-weighted new water fractions for all time steps, $^{Q}F_{new}^{*}$). These results illustrate that the numerical values of new water fractions, and also their meaning, are intrinsically tied to the sampling frequency: "new" water is water that fell as rain during the last sampling
interval, whether that interval is 7 hours or 7 days. The small new water fraction obtained from the 7-hourly data is not surprising, as 7 hours is a relatively short time for any raindrop to reach the catchment outlet, unless it lands directly in the channel itself. Instead, most streamflow is dominated by older water that originated from previous precipitation events and has been stored within the catchment for months or longer. Unsurprisingly, across all sites and sampling frequencies, volume-weighted new water fractions were larger than unweighted new water fractions, because volume-weighting gives more
emphasis to higher flows which typically contain larger proportions of recent precipitation.

In contrast to the pronounced differences between the new water fractions determined from the 7-hourly and weekly time series, the weekly sampling at Lower Hafren and Tanllwyth yielded broadly similar new water fractions, with slightly higher values at Tanllwyth than at Lower Hafren. The higher new water fractions in the Tanllwyth catchment can be plausibly
attributed to its higher prevalence of low-permeability gley soils (Neal et al., 2004b), which would tend to promote faster near-surface flows.

Event new water fractions $\left(^{Qp}F_{new}\right)$ are calculated only over time steps with precipitation, and thus are always larger than new water fractions averaged over all time steps, including rainless periods $\left(^{Q}F_{new}\right)$. In the weekly data, this difference was
relatively small, because almost all time steps had precipitation (roughly 70% of weeks had precipitation rates above the threshold, here set to $\bar{P} \approx 0.1$ mm h$^{-1}$ for both weekly and 7-hourly sampling). In the 7-hourly data, by contrast, the event new





water fraction $\left(^{Q_p}F_{new}\right)$ was roughly 1.5-3 times higher than the new water fraction for all time steps $\left(^{Q}F_{new}\right)$, because most 7-hour intervals were rainless (and therefore could not contribute any new water, because new water is defined as precipitation that fell within the current time step). Only about 35% of 7-hourly periods had precipitation rates higher than the threshold; roughly 50% had no precipitation at all, and another 15% had some precipitation but less than the threshold. New water

fractions of precipitation ($^{P}F_{new}$, the fractions of precipitation becoming streamflow in the same time step) were somewhat smaller than event new water fractions ($^{Q_p}F_{new}$, the fractions of streamflow originating as precipitation in the same time step). This was because during most storms the rainfall rate will be higher than the streamflow rate, so the ratio between same-time-step streamflow and the total rainfall rate $\left(^{P}F_{new}\right)$ will necessarily be smaller than the ratio between same-time-step streamflow and the total streamflow rate $\left(^{Q_p}F_{new}\right)$ (Kirchner, 2019). This contrast in water fluxes was less pronounced at the

weekly time scale, so the contrast between the new water fractions of precipitation and discharge was also less pronounced. The volume-weighted new water fractions of precipitation and discharge $\left(^{P}F_{new}^{*} \text{ and } ^{Q}F_{new}^{*}\right)$ are related by the ratio of total discharge to total precipitation; in Plynlimon's very humid climate, this ratio is close to 1 and thus these two new water fractions were nearly equivalent.

The previous paragraphs, along with Table 1, demonstrate that substantially different values can be obtained, depending on which type of new water fraction is calculated. Which new water fractions are the correct ones will depend on the scientific question, because they provide somewhat different types of information. For this reason, it may be beneficial to compute several different variants, as we have done here, to obtain a more holistic picture of catchment processes. In general, one can expect that volume-weighted new water fractions will give more reproducible results than unweighted new water fractions,

because they will give less weight to low-volume samples that may have anomalous tracer values.

New water fractions determined from deuterium and oxygen-18 agreed within one pooled standard error. New water fractions determined from the dry-deposition-filtered 7-hourly chloride time series were systematically smaller than those calculated from stable isotopes, but nonetheless in a similar range. This result suggests that the dry deposition filtering performed on the

7-hourly chloride data worked reasonably well. Conversely, new water fractions determined from weekly chloride samples (which were not corrected for effects of dry deposition) were significantly larger than those obtained from stable water isotopes.





## 5.2 Effect of the precipitation threshold

We calculated new water fractions assuming different precipitation thresholds, below which samples were considered unreliable and excluded from the analysis. Volume-weighted event new water fractions $\left(^{Q_p}F_{new}^*\right)$ increased with increasing precipitation thresholds in our analysis (Fig. 8a,b), as time steps with higher volumes were inherently given increasing weight.

Conversely, volume-weighted new water fractions determined for all time steps $\left(^{Q}F_{new}^*\right)$ were affected less by the precipitation threshold, due to the scaling factor accounting for the increasing fraction of days without precipitation (Fig. 8c,d).

The change in new water fractions with increasing precipitation thresholds was very similar between the two stable water isotopes (see Fig. 8). In contrast, new water fractions calculated directly from the 7-hourly chloride time series (without

filtering for dry deposition effects) were substantially smaller. This was likely due to the higher variability in precipitation concentrations due to dry deposition effects, leading to a stronger apparent damping of chloride concentrations than stable water isotopes between precipitation and streamwater. After precipitation chloride values that were potentially affected by dry deposition were removed from the analysis (6.6% of samples, equaling 50 out of 751 data points), new water fractions determined from chloride agreed within error with those determined from the stable water isotopes, as long as the precipitation

threshold was higher than the mean precipitation rate (Fig. 8).

For the weekly chloride data, we expected the effect of dry deposition to be less important (and more difficult to assess), so we only performed a general outlier removal. The resulting chloride time series yielded substantially higher new water fractions than those obtained from the stable water isotopes. This discrepancy was largest for small precipitation thresholds, but

substantial throughout. This effect cannot be explained by the dry deposition of chloride alone, because we would expect this to increase the variability in the precipitation time series and thus reduce the new water fraction (as we saw in the 7-hourly samples). Instead, the damping from precipitation to streamflow seems to be weaker for chloride than the stable water isotopes in the weekly data set. This was previously discussed by Neal and Rosier (1990), who linked a lower degree of damping for chloride to evaporative concentration of chloride in the catchment, resulting in a more heterogeneous distribution of chloride.

Recommending an ideal precipitation threshold is not trivial and likely depends on the frequency and intensity of rain events, as well as the sampling frequency. If the precipitation threshold is set too low, potentially unreliable data points from small-volume samples will be included in the analysis. Conversely, if the threshold is set too high, many samples will be excluded from the analysis, increasing the uncertainty in the calculated new water fractions due to the reduced sample size. In this study,

a precipitation threshold of 0.1 mm h$^{-1}$ for both 7-hourly and weekly data gave, in our view, reasonable results. This threshold led to an exclusion of approximately 2% and 5% of the total precipitation volumes, and of approximately 16% and 21% of the isotope samples, for 7-hourly and weekly sampling, respectively. This threshold value was used in all analyses presented here, unless explicitly stated otherwise.





## 5.3 Comparison of results from aggregated 7-hourly samples and weekly samples

To test how different sampling frequencies could affect estimates of new water fractions, we aggregated the 7-hourly data to synthesize longer sampling intervals. As the length of the sampling interval changes, so does both the magnitude, and the meaning, of the new water fraction. "New" water is defined as streamflow that fell as precipitation within the last sampling

interval. Thus it is not surprising, for example, that the fraction of streamflow that fell as precipitation within the last week (the new water fraction for weekly sampling) will be larger than the fraction of streamflow that fell as precipitation within the 7 hours (the new water fraction for 7-hourly sampling). As expected, the new water fraction increased with the length of the (synthetic) sampling intervals shown in Fig. 9. The curves shown here increase rather steeply over the first day, and more gradually over longer sampling intervals. The new water fraction obtained from synthetic weekly sampling was around three

times higher than from 7-hourly sampling in case of the stable isotopes. One might have expected an increase by a factor of 24 between sampling interval lengths of seven hours to one week (because 24 x 7 h = 1 week). The observed increase is smaller for two reasons. The first reason is that, as the sampling interval increases, the average time lag also increases between any individual precipitation event and the moment that the stream is sampled. As this lag time increases, so does the chance that recent precipitation will have already passed the sampling point by the time that the sample is collected. The second reason

arises from how stochastic rainfall events are aggregated as the sampling interval increases. In any given week (for example), there may be several 7-hour periods when rain falls, but many others when it does not. Event new water fractions of the 7-hourly data would count only the 7-hour periods with rain, and would ignore the rest. The event new water fraction for the entire week will include these rainless periods (but the week will still be classified as an "event" because it includes some periods of rainfall). Thus the event new water fraction at the weekly timescale will be smaller than if it consisted entirely of

rainy intervals. (Of course, even 7-hour events may include rainless periods, so although this thought experiment explains how new water fractions scale with the sampling interval, it does not argue for any particular interval being the "correct" one.)

In Fig. 9, similar to Table 1 and Fig. 8, the new water fractions determined from deuterium and oxygen-18 were nearly identical, whereas those determined from chloride deviated somewhat. If dry deposition was not accounted for, the new water fraction

was substantially smaller than the new water fraction determined from stable isotopes, particularly for sampling intervals of one day or less. If the effect of dry deposition was filtered out, new water fractions determined from chloride were within one standard error of those determined from stable water isotopes, except for the shortest sampling interval. This pattern was consistent for both unweighted and volume-weighted new water fractions.

We can directly compare the new water fractions of all three sites, if we put them on a consistent time base. We aggregated the 7-hourly samples at Upper Hafren to mimic the samples that would have been obtained through weekly sampling. We then took a subset of the weekly data at Lower Hafren and Tanllwyth, coinciding with the period of the 7-hour sampling at Upper Hafren (and excluding the long data gap in the 7-hour samples between December 2007 and March 2008). With all three





catchments on this consistent time base, their unweighted new water fractions were broadly similar, with a small increase in new water fractions from Lower Hafren to Tanllwyth to Upper Hafren. Their volume-weighted new water fractions, however, were systematically larger at Lower Hafren and Tanllwyth than at Upper Hafren, which is consistent with faster routing of new water, particularly at high flows, due to the drainage ditches in the Lower Hafren and Tanllwyth plantation forests.

## 5.4 Variation in new water fractions with hydraulic regime and season

We calculated new water fractions from the oxygen-18 time series, subsampled to capture different percentiles of the discharge and precipitation distributions (Fig. 10). The new water fraction increased with discharge and precipitation, indicating (unsurprisingly) that recent precipitation contributed more to streamflow during large events. This was likely due to greater

saturation of soils during intense rain events, resulting in a greater dominance of shallow flowpaths and thus promoting faster transport of precipitation to the stream. In addition, channel networks (Godsey and Kirchner, 2014; Zimmer and McGlynn, 2017) and near-stream saturated zones (Dunne et al., 1991) expand with increasing precipitation, causing raindrops to fall closer to the channel and therefore reach the catchment outlet faster (van Meerveld et al., 2019).

Event new water fractions for the highest 10% of discharge were somewhat larger than the volume-weighted means (dashed lines in Fig. 10a-c) and about three times the unweighted means (solid lines in Fig. 10a-c). During these wet conditions, recent (same-week) precipitation accounted for roughly 25-30% of streamflow in the weekly samples, and recent (same-7-hours) precipitation accounted for roughly 6% of streamflow in the 7-hour samples. Conversely, event new water fractions for the lowest 40% of the discharge distributions were typically about half, or less, of the unweighted means. For small water fluxes,

new water fractions dropped to less than 5% in case of weekly time steps, and close to 0% for 7-hourly sampling. (In the top panels of Fig. 10, in contrast to the rest of this paper, we used a precipitation threshold of 0 mm h$^{-1}$ when calculating event new water fractions, because a higher precipitation threshold would have excluded most of the low-discharge samples, which mostly coincide with very low precipitation rates. Thus we needed to eliminate the precipitation threshold, to reveal how new water fractions vary across the entire discharge range.) New water fractions of precipitation were also highest during the most

intense rain events, and smaller at low precipitation rates. They were always lower than event new water fractions estimated for similar water fluxes, for the reasons outlined in Sect. 5.1.

Volume-weighted event new water fractions were visibly higher in fall and winter compared to spring and summer, and this pattern was mirrored, although less distinctly, in unweighted event new water fractions as well (Fig. 10d-f). This pattern was

similar for 7-hourly and weekly sampling, and also for all three catchments. The higher new water fractions during fall and winter may be attributable to higher precipitation and lower evapotranspiration during these months, even though the climate at Plynlimon is generally humid throughout the year, with only slight seasonal differences in precipitation (summer rain





accounts for 40 % of annual rainfall, Kirby et al., 1991). A catchment with stronger seasonality in rainfall could potentially exhibit an even more pronounced seasonal pattern in new water fractions.

## 5.5 Transit time distributions

5   We estimated transit time distributions by ensemble hydrograph separation based on weekly and 7-hourly sampling of oxygen-18. Both transit time distributions of discharge ("backward transit time distributions") and transit time distributions of precipitation ("forward transit time distributions") were low and broad, decreasing gradually at greater lag times, when calculated over all data (solid colored symbols in Fig. 11). Volume-weighted transit time distributions (open gray symbols in Fig. 11) showed somewhat stronger peaks at short lag times, consistent with transport being faster during larger events.

Calculating transit time distributions separately for different seasons, we found a less damped response in fall/winter, compared to spring/summer (Fig. 12). This is consistent with the observation that new water fractions tend to be higher in the colder months (see Fig. 10), possibly due to higher rainfall and lower evapotranspiration, and therefore wetter catchment conditions and higher streamflow, during these months. The seasonal differences between the transit time distributions largely 15  disappeared at lag times longer than about 1-1.5 days. This observation further highlights the likely role of wetter catchment conditions in promoting faster transport of rainwater to the stream during the fall/winter.

## 5.6 Comparison with spectral estimates of transit time distributions

As described in Sect. 4.2, transit time distributions can also be estimated from the power spectra of the tracer time series. The 20  gamma model, when multiplied by the 7-hourly precipitation tracer power spectrum, fitted the streamwater tracer power spectrum closely at timescales of less than roughly one month, corresponding to frequencies above roughly 10 per year (Fig. 13a,b). The fitted gamma parameters yielded transit time distributions that corresponded closely to those estimated from ensemble hydrograph separation (Fig. 13c,d). This result is noteworthy, because although both estimation methods obviously relied on the same source data, they involve different mathematical procedures and different underlying assumptions. For 25  example, the spectral fitting method assumed that transit times are gamma-distributed; by contrast, ensemble hydrograph separation makes no assumption about the shape of the transit time distribution, but nonetheless yielded results that are broadly consistent with a gamma distribution. Fig. 13 does not provide a strong constraint on the shape of the distribution on timescales much shorter than 7 hours or longer than 7 days. Nonetheless, the similarities between the distributions obtained by spectral fitting and ensemble hydrograph separation strengthen our confidence that both methods can reliably quantify the transit time 30  behavior of real-world catchments. These similarities are not limited to the 7-hourly data shown in Fig. 13; they are also seen, although with greater uncertainties, in the transit time distributions obtained from weekly data at Lower Hafren and Tanllwyth (supplemental figures S6 – S7).



The spectral fitting method assumes that the transit time distribution is time-invariant (i.e., stationary). In theory the regression techniques underlying ensemble hydrograph separation make the same assumption, but the benchmark tests of Kirchner (2019) show that they nonetheless reliably estimate the ensemble averages of nonstationary transit time distributions. Figure 13c,d
therefore suggests that the spectral fitting method also yields ensemble averages of nonstationary transit time distributions, but this should be verified using benchmark tests.

The transit time distributions of the two isotopes appeared very similar, both over timescales of days (Fig 13c,d) and weeks (Supplementary Figures S6-S7). Perhaps surprisingly, however, the gamma distributions fitted to the spectra of the two
isotopes can yield markedly different estimates of mean transit time (Supplementary Table S1). For example, the gamma distribution derived from deuterium in Fig. 13c implied a mean transit time of 0.63±0.06 years, but the gamma distribution derived from oxygen-18 implied a mean transit time that was 35-fold longer (22.2±4.6 years). The discrepancy was smaller, but still substantial, for the volume-weighted distributions shown in Fig. 13d (0.081±0.006 versus 0.140±0.011 years for deuterium and oxygen-18, respectively). Across all sites and sampling frequencies, we found that the fitted shape factors $k$
were smaller, the fitted scale factors $\theta$ were larger, and the resulting mean transit times $\bar{\tau} = k\theta$ were longer, when derived from oxygen-18 than from deuterium, with mean transit times typically differing by roughly a factor of two. Despite the similarities in the short-time behavior of the transit time distributions shown here, their mean transit times are largely determined by their long-time behavior, which is poorly constrained by convolution methods, including the spectral fitting technique used here (and is not estimated at all by ensemble hydrograph separation). Our analysis thus reinforces earlier
concerns regarding mean transit times estimated from stable isotope tracers (Stewart et al., 2010; Seeger and Weiler, 2014; Kirchner, 2019), even when, as here, the transit time distribution itself can be reliably estimated over a shorter range of lag times.

## 6 Summary and conclusions

This study represents the first attempt to assess transport and mixing processes in a real-world catchment using ensemble hydrograph separation. Using this approach, we quantified the contribution of recent precipitation to streamflow in three catchments at Plynlimon, Wales, based on 7-hourly and weekly time series of stable water isotopes. The weekly time series revealed that, on average, roughly 13-15% of streamwater consisted of precipitation that fell within the previous week, whereas this "new water fraction" decreased to roughly 3% for 7-hour time steps (Table 1). This illustrates that both the numerical
value and meaning of "new" water are intrinsically tied to the sampling frequency, because "new" water is defined as streamflow that fell as precipitation during the previous time step.



Our analyses show that the streamflow and precipitation rates strongly influenced the amount of recent precipitation found in streamflow. Larger events yielded larger new water fractions (Fig. 10), indicating that the catchment is more connected during wet conditions, with precipitation inputs being transmitted faster to the catchment outlet. As a consequence, volume-weighted transit time distributions were systematically steeper than unweighted transit time distributions (Fig. 11). Seasonal variations in water fluxes also shaped seasonal patterns of new water contributions to streamflow; new water fractions were higher (Fig. 10), and transit time distributions were steeper (Fig. 12), in the fall/winter months when precipitation is high and evapotranspiration is low, leading to wetter catchment conditions and higher streamflow. These results highlight that the transport of water through catchments is not determined by catchment characteristics alone, but instead by the interaction between catchment characteristics and climatic conditions.

Overall, however, we observed relatively small amounts of recent precipitation in streamflow at Plynlimon, indicating that there is substantial residual storage even when the catchment is relatively dry. This residual storage mixes with most of the incoming precipitation, damping its tracer fluctuations. The catchment retains its chemical and isotopic memory because the volume of incoming water is small compared to the substantially larger residual storage. In contrast, only a small fraction of precipitation is transmitted rapidly enough to streamflow that it retains its chemical and isotopic signature.

Transit time distributions estimated from oxygen-18 and deuterium agreed closely with one another, whether calculated by ensemble hydrograph separation or by power spectrum fitting. We also found good agreement between new water fractions calculated from the oxygen-18 and deuterium time series (Tables 1 and 2, Fig. 8 and 9). By aggregating the 7-hourly samples to weekly frequency, we could also show that the differences in new water fractions among the three catchments (Table 2) were consistent with differences in their soil characteristics and in the prevalence of drainage ditches associated with plantation forestry. Together, these findings demonstrate the reliability and utility of the stable water isotope data.

In contrast, new water fractions determined from weekly time series of chloride concentrations were significantly larger than those obtained from time series of stable water isotopes (Table 1, Fig. 8). This may be linked to spatially and temporally variable effects of evapoconcentration and dry deposition of chloride, but the exact extent of these effects is difficult to quantify because all weekly precipitation samples were probably affected to some extent. Identification and removal of dry-deposition-affected samples was easier in the 7-hourly chloride data, and resulted in new water fractions that more closely resembled those derived from stable water isotopes (Table 1, Fig. 8 and 9). Substantial differences still remained, however, and we conclude that stable water isotopes provide a more reliable basis for quantifying catchment transport timescales.

The stable isotope measurements presented here cover periods of several months to years at 7-hourly and weekly frequencies, making them some of the longest and most detailed publicly available catchment isotope data sets. They thus provide an opportunity to investigate catchment transport and mixing in great detail, and the analyses presented here can be considered





as just a starting point for further work. Moreover, extensive solute data sets are already publicly available for the same sites and sampling periods (Neal et al., 2013b, c; Norris et al., 2017). The data sets of stable water isotopes presented here thus complement the already available data, likely enabling many future analyses of catchment behavior, particularly with respect to catchment-scale reaction processes.

**Acknowledgements**

The data sets described in this manuscript are attached as supplemental information. The data sets will also be archived in the EnviDat database of WSL/ETH Zurich upon acceptance of this manuscript. The authors thank the Plynlimon field staff at the Centre for Ecology and Hydrology (CEH) for their contributions to this work, and Daniele Pezzotta and Stefan Weber for their
10    contributions to the isotopic analyses. We also acknowledge CEH's long-term financial support of the Plynlimon hydrochemistry study, and WSL's support for the isotopic analyses. The first author received an ETH Zurich Postdoctoral Fellowship partly funded by the European Union under the 7th Framework Programme.



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





**Table 1: New water fractions (± standard errors) calculated from 7-hourly and weekly measurements of deuterium, oxygen-18, and chloride (chloride time series at 7-hourly resolution were corrected for dry deposition before the calculation of new water fractions). From top to bottom: new water fractions calculated using all time points, event new water fractions calculated from time steps with precipitation, and new water fractions of precipitation, each calculated with and without volume-weighting. New water fractions cannot be directly compared between the two sampling frequencies.**

|  | 7-hourly sampling at Upper Hafren 2007-2009 | Weekly sampling at Lower Hafren 2005-2009 | Weekly sampling at Tanllwyth 2005-2009 |
|---|---|---|---|
| New water fractions for all time steps[a], $^Q F_{new}$ (%): |  |  |  |
| Deuterium | $0.88 \pm 0.11$ | $4.93 \pm 0.72$ | $5.58 \pm 0.71$ |
| Oxygen-18 | $0.90 \pm 0.15$ | $4.43 \pm 0.87$ | $5.69 \pm 0.82$ |
| Chloride | $0.54 \pm 0.11$ | $11.07 \pm 2.12$ | $11.93 \pm 2.02$ |
| Volume-weighted new water fractions for all time steps[b], $^Q F_{new}^*$ (%): |  |  |  |
| Deuterium | $2.95 \pm 0.41$ | $13.19 \pm 1.88$ | $14.61 \pm 1.81$ |
| Oxygen-18 | $2.90 \pm 0.49$ | $13.26 \pm 2.23$ | $15.27 \pm 2.09$ |
| Chloride | $1.99 \pm 0.41$ | $27.40 \pm 4.64$ | $24.25 \pm 5.54$ |
| Event new water fractions[c], $^{Qp} F_{new}$ (%): |  |  |  |
| Deuterium | $2.42 \pm 0.31$ | $7.36 \pm 1.08$ | $8.33 \pm 1.07$ |
| Oxygen-18 | $2.47 \pm 0.40$ | $6.61 \pm 1.30$ | $8.50 \pm 1.23$ |
| Chloride | $1.49 \pm 0.31$ | $16.54 \pm 3.16$ | $17.82 \pm 3.01$ |
| Volume-weighted event new water fractions[d], $^{Qp} F_{new}^*$ (%): |  |  |  |
| Deuterium | $5.02 \pm 0.70$ | $14.57 \pm 2.08$ | $15.90 \pm 1.97$ |
| Oxygen-18 | $4.93 \pm 0.83$ | $14.63 \pm 2.46$ | $16.62 \pm 2.28$ |
| Chloride | $3.39 \pm 0.70$ | $30.25 \pm 5.13$ | $26.40 \pm 6.03$ |
| New water fractions of precipitation[e], $^P F_{new}$ (%): |  |  |  |
| Deuterium | $1.41 \pm 0.18$ | $5.93 \pm 0.87$ | $7.33 \pm 0.94$ |
| Oxygen-18 | $1.44 \pm 0.23$ | $5.32 \pm 1.05$ | $7.48 \pm 1.08$ |
| Chloride | $0.87 \pm 0.18$ | $13.32 \pm 2.54$ | $15.68 \pm 2.65$ |
| Volume-weighted new water fractions of precipitation[f], $^P F_{new}^*$ (%): |  |  |  |
| Deuterium | $2.86 \pm 0.40$ | $11.16 \pm 1.59$ | $13.30 \pm 1.64$ |
| Oxygen-18 | $2.81 \pm 0.47$ | $11.21 \pm 1.88$ | $13.91 \pm 1.90$ |
| Chloride | $1.93 \pm 0.40$ | $23.17 \pm 3.93$ | $22.08 \pm 5.04$ |

[a] Following Eq. 14 in (Kirchner, 2019)
[b] Following Eq. 18 in (Kirchner, 2019)
[c] Following Eq. 10 in (Kirchner, 2019)
[d] Following Eq. 18 in (Kirchner, 2019)
[e] Following Eq. 21 in (Kirchner, 2019)

[f] Following Eq. 28 in (Kirchner, 2019), thus calculated through a rescaling of $^{Qp} F_{new}^*$.





**Table 2: Comparison of new water fractions (± standard errors) from weekly data at the Upper Hafren, Lower Hafren and Tanllwyth.**

|  | Upper Hafren (from aggregated 7-hourly sampling) 2007-2009[a] | Lower Hafren (from weekly sampling) 2007-2009[b] | Tanllwyth (from weekly sampling) 2007-2009[b] |
|---|---|---|---|
| Unweighted new water fractions for all time steps, $^{Q}F_{new}$ (%): | | | |
| deuterium | $5.28 \pm 1.01$ | $3.62 \pm 0.75$ | $4.21 \pm 0.84$ |
| oxygen-18 | $4.79 \pm 1.11$ | $3.72 \pm 0.90$ | $4.46 \pm 1.01$ |
| Volume-weighted new water fractions for all time steps, $^{Q}F_{new}^{*}$ (%): | | | |
| deuterium | $10.88 \pm 1.85$ | $14.92 \pm 3.03$ | $15.18 \pm 2.66$ |
| oxygen-18 | $9.06 \pm 2.02$ | $16.32 \pm 3.53$ | $16.28 \pm 2.87$ |
| Unweighted event new water fractions, $^{Qp}F_{new}$ (%): | | | |
| deuterium | $6.99 \pm 1.34$ | $5.20 \pm 1.07$ | $6.05 \pm 1.20$ |
| oxygen-18 | $6.34 \pm 1.47$ | $5.35 \pm 1.30$ | $6.41 \pm 1.45$ |
| Volume-weighted event new water fractions, $^{Qp}F_{new}^{*}$ (%): | | | |
| deuterium | $11.87 \pm 2.02$ | $16.26 \pm 3.30$ | $16.27 \pm 2.85$ |
| oxygen-18 | $9.88 \pm 2.20$ | $17.78 \pm 3.85$ | $17.45 \pm 3.08$ |
| Unweighted new water fraction of precipitation, $^{P}F_{new}$ (%): | | | |
| deuterium | $6.14 \pm 1.18$ | $4.26 \pm 0.88$ | $5.44 \pm 1.08$ |
| oxygen-18 | $5.56 \pm 1.29$ | $4.38 \pm 1.06$ | $5.76 \pm 1.30$ |
| Volume-weighted new water fraction of precipitation, $^{P}F_{new}^{*}$ (%): | | | |
| deuterium | $10.14 \pm 1.72$ | $12.74 \pm 2.59$ | $13.96 \pm 2.44$ |
| oxygen-18 | $8.44 \pm 1.88$ | $13.93 \pm 3.01$ | $14.98 \pm 2.64$ |

[a] A synthetic weekly data set for Upper Hafren was created by taking weekly volume-weighted averages of 7-hourly precipitation, and weekly subsamples of 7-hourly Upper Hafren streamwater, at dates and times corresponding to the regular weekly sampling at Lower Hafren and Tanllwyth.

[b] The weekly data sets for Lower Hafren and Tanllwyth were shortened to the period coinciding with the 7-hour sampling at Upper Hafren (also omitting the sampling gap between December 2007 and March 2008).





**Figure 1: The Headwater catchments of the rivers Severn and Wye at Plynlimon, Wales. Stable water isotopes in precipitation were recorded at Carreg Wen (gray circle 1, 575 m a.s.l.) at weekly and 7-hourly resolution. Weekly streamwater samples of stable water isotopes were collected at Lower Hafren and Tanllwyth (gauging stations indicated by triangles 1 and 2, at 356 m a.s.l. and 352 m a.s.l., respectively), and at 7-hourly resolution at Upper Hafren (gauging station denoted by triangle 3, 550 m a.s.l.). The weather stations at Carreg Wen and Tanllwyth (gray circles 1 and 2) were used to record rates of precipitation at 7-hourly and weekly intervals, respectively.**





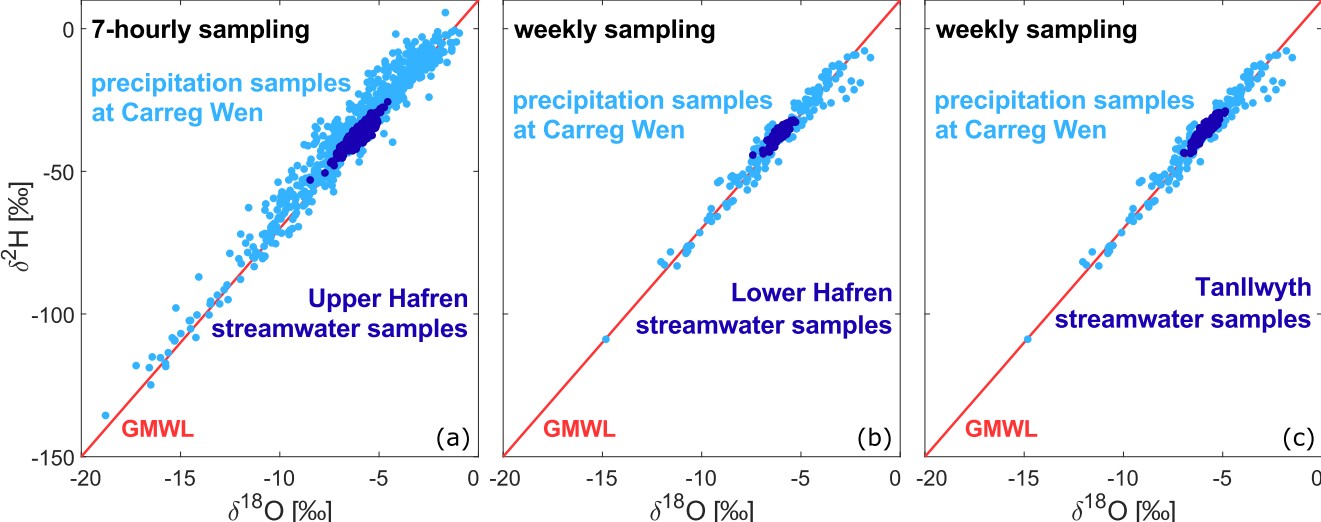

**Figure 2: Dual-isotope plots for precipitation (light blue) and streamwater (dark blue) samples from 7-hourly (a) and weekly sampling (b-c). Both precipitation and streamwater samples generally fall close to the global meteoric water line (GMWL, red line), and thus show little evidence of evaporative fractionation.**



**Figure 3: Seasonal variations in deuterium excess for 7-hourly and weekly precipitation samples (a-b) and streamwater samples (c-d). In each boxplot, the center line indicates the median, whereas the box delimits the 25th and 75th percentiles. Whiskers extend twice the interquartile range, or to the maximum/minimum of the data. Outliers beyond the whiskers are indicated by separate points. Axis scales differ between panels (a,b) and (c,d), reflecting the greater variability in precipitation deuterium excess. There are small but distinct seasonal differences in deuterium excess in both precipitation and streamwater. However the deuterium excess patterns in the 7-hourly streamwater samples mirror those in the weekly grab samples (which were not vulnerable to evaporation in the field), suggesting that these patterns reflect real-world seasonal variations in deuterium excess, and that any evaporative fractionation effects of storage in the 7-hourly samples are small.**





**Figure 4: Variations in deuterium excess in 7-hourly samples as a function of the length of time that they were stored inside the field autosamplers. Boxplots are defined as described in Fig. 3. Axis scales differ between panels (a,b) and (c,d), reflecting the greater variability in precipitation deuterium excess. There is no systematic effect of storage duration on deuterium excess during either summer (May-October, panels a,c) or winter (November-April, panels b,d), indicating that any evaporative fractionation during storage was negligible. Deuterium excess in streamwater samples is systematically lower during summer, likely reflecting real-world seasonal variations in deuterium excess (see also Fig. 3).**

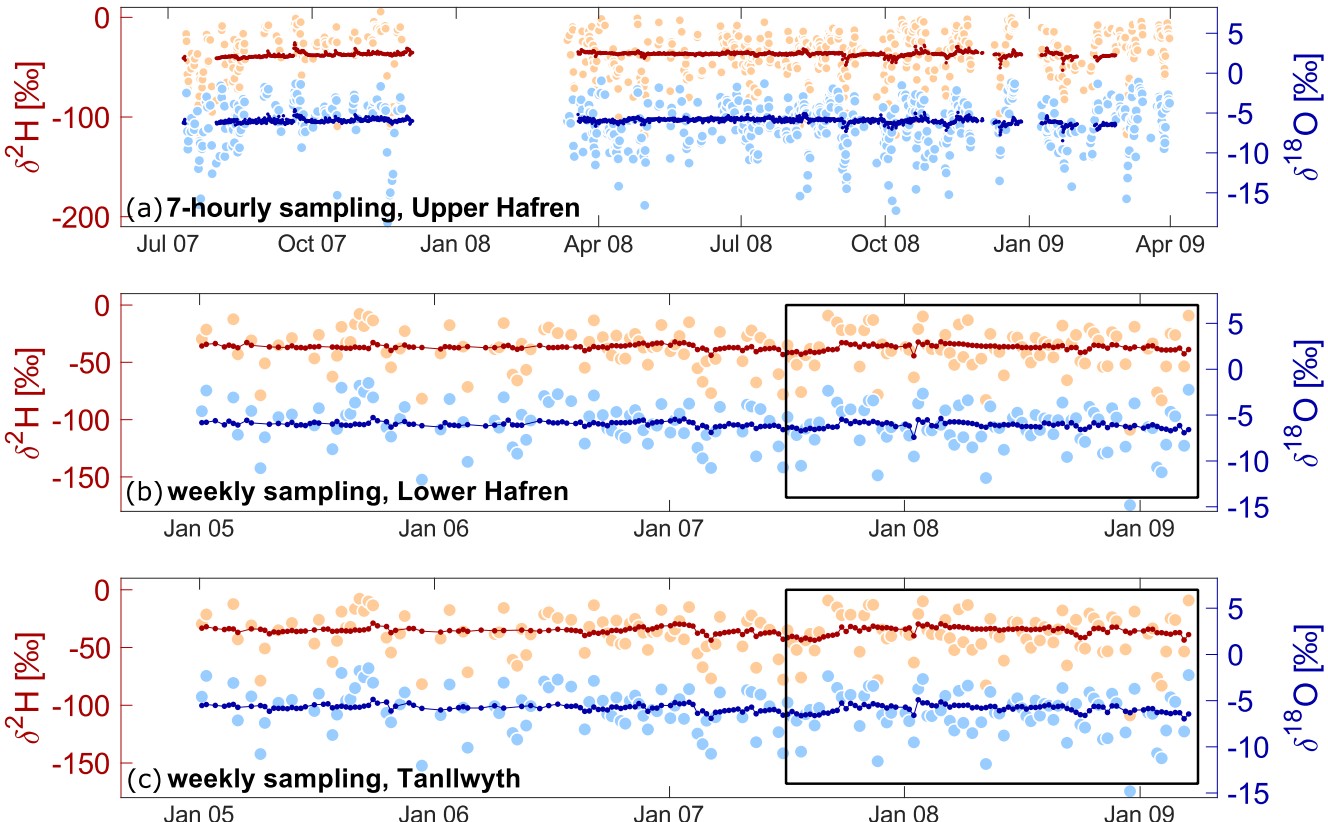

**Figure 5: Time series of stable water isotopes measured at 7-hourly resolution (a) and weekly resolution (b-c) in precipitation (lighter colors) and streamwater (darker colors). All precipitation samples were collected at Carreg Wen, and 7-hourly streamwater samples were collected at the Upper Hafren catchment outlet, whereas weekly streamwater samples were collected at the Lower Hafren and Tanllwyth catchment outlets. The deuterium axis (left) is compressed by a factor of eight relative to the oxygen-18 axis (right). Consequently, fluctuations along the meteoric water line would appear equally large for both isotopes. The black rectangles shown in the weekly plots indicate the period of the 7-hourly sampling.**



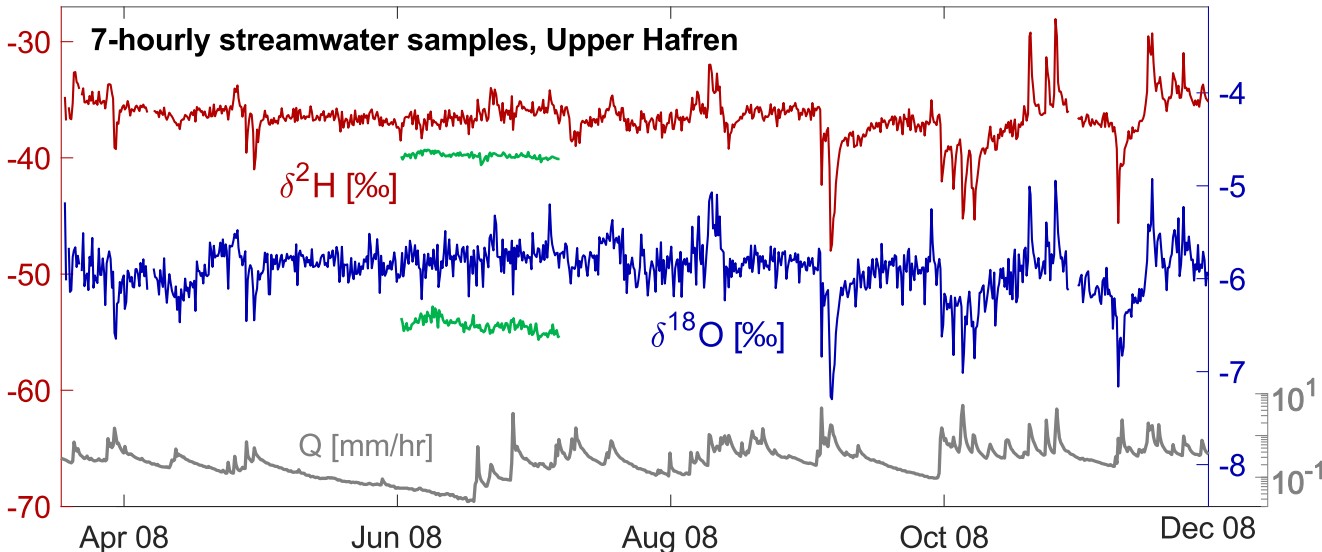

**Figure 6: A close-up of the 8 months of nearly complete streamwater samples from 7-hourly sampling. Deuterium (left axis) is shown in red, oxygen-18 (right axis) in blue, and discharge is shown in gray. The fluctuations of the replicate QC standards from the reproducibility test (green) illustrate the magnitude of the variability between samples due to analytical noise.**



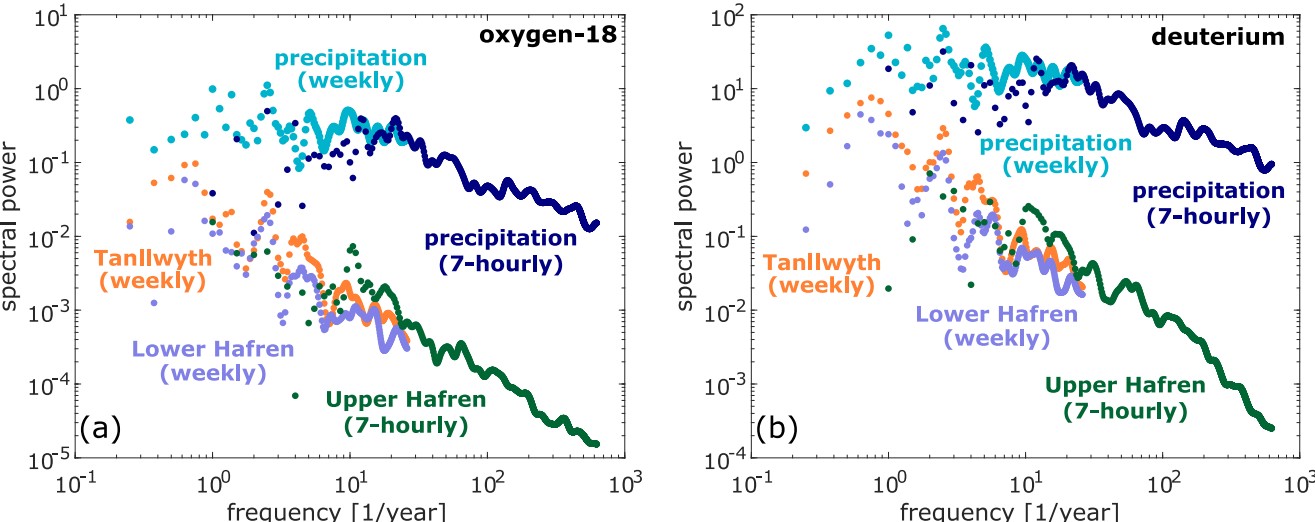

**Figure 7: Power spectra of fluctuations in oxygen-18 (a) and deuterium (b) in precipitation and streamwater, calculated using the weighted wavelet and alias filtering methods of Kirchner (2005) and Kirchner and Neal (2013). Weekly and 7-hourly streamwater spectra are not strictly comparable because they are measured in different streams, whereas weekly and 7-hourly precipitation spectra were calculated from samples collected at the same location (although during partly non-overlapping time periods).**





**Figure 8: Effects of different precipitation thresholds on volume-weighted new water fractions for 7-hourly sampling at Upper Hafren (a, c) and weekly sampling at Lower Hafren (b, d). Error bars indicate one standard error. (a)-(b) show volume-weighted event new water fractions $\left(^{Q_p}F_{new}^*\right)$, which consider only time steps with precipitation. Higher precipitation thresholds lead to higher new water fractions in these panels, because time steps with low precipitation rates are usually associated with lower discharges. As these are excluded, fewer and fewer low-flow time steps are considered, increasing the relative importance of time steps with high discharge, which also tend to have higher new water fractions. Conversely, the precipitation threshold has less effect on volume-weighted new water fractions for all time steps ($^{Q}F_{new}^*$, c-d), because these include a factor that accounts for the fraction of days without precipitation. Precipitation thresholds have been slightly jittered for better visibility of the different tracers.**




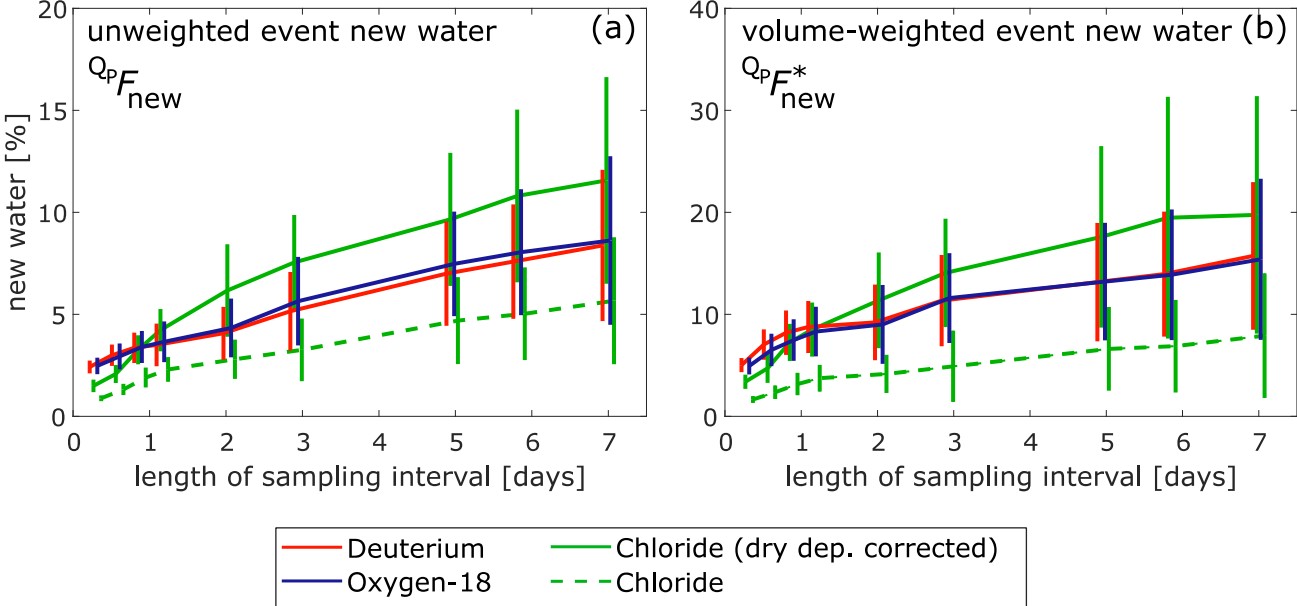

**Figure 9:** Effects of aggregating 7-hourly data to longer sampling intervals for deuterium (red), oxygen-18 (blue), and chloride (green) with and without correction for dry deposition effects. Error bars indicate one standard error. The new water fractions determined from the stable water isotopes are nearly identical, whether unweighted (a) or volume-weighted (b). Sampling interval lengths have been slightly jittered for better visibility of the different tracers.



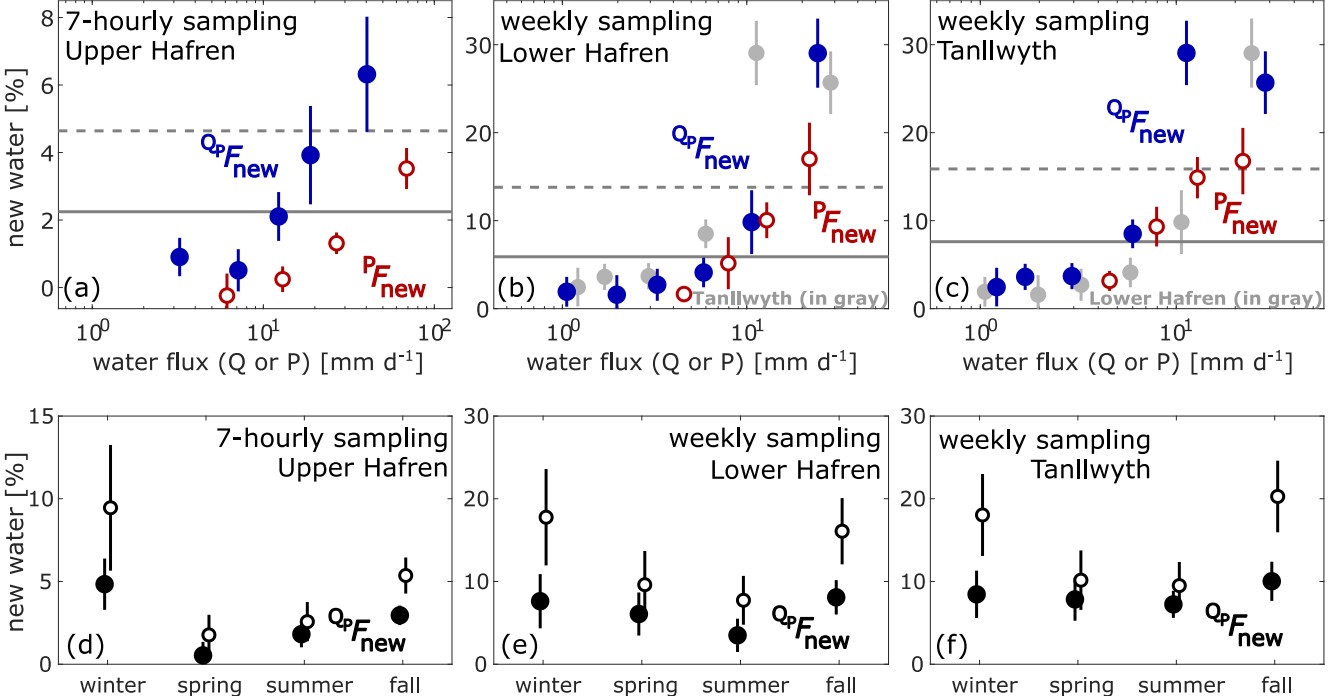

**Figure 10:** Event new water fractions $\left(^{Q_P}F_{new}\right)$ and new water fractions of precipitation $\left(^{P}F_{new}\right)$ for different discharge rates, precipitation rates, and seasons, calculated from time series of oxygen-18. Error bars indicate one standard error. In panels (a)-(c), solid circles indicate event new water fractions, plotted as functions of discharge rates, and open circles indicate new water fractions of precipitation, plotted as functions of precipitation rates. Gray markers in the background of panel (b) show event new water fractions from panel (c), and vice versa, to facilitate comparison between Lower Hafren and Tanllwyth. Event new water fractions are calculated for different percentiles of the discharge regime (in blue, 0–30, 30–60, 60–80, 80–90, and 90–100 for 7-hourly sampling, and 0–20, 20–40, 40–60, 60–80, 80–90, and 90–100 for weekly sampling; percentiles are calculated based only on discharge of time steps with precipitation, and precipitation thresholds were set to 0 mm h⁻¹). New water fractions of precipitation are calculated for different percentiles of precipitation rates above the precipitation threshold of 0.1 mm h⁻¹ (in red, 0-40, 40-60, 60-85, 85-100). Solid and dashed gray lines indicate unweighted and volume-weighted event new water fractions, respectively, across all discharge values. Panels (d)-(f) show event new water fractions, across all discharge values, determined separately for winter (December–February), spring (March–May), summer (June–August) and fall (September–November). Open and solid circles indicate volume-weighted and unweighted event new water fractions, respectively, in panels (d)-(f).





**Figure 11: Transit time distributions of discharge («backward» TTDs, left) and precipitation («forward» TTDs, right) calculated from 7-hourly (a, b) and weekly (c-f) time series of oxygen-18. Solid circles indicate unweighted transit time distributions, whereas open gray circles indicate volume-weighted transit time distributions.**





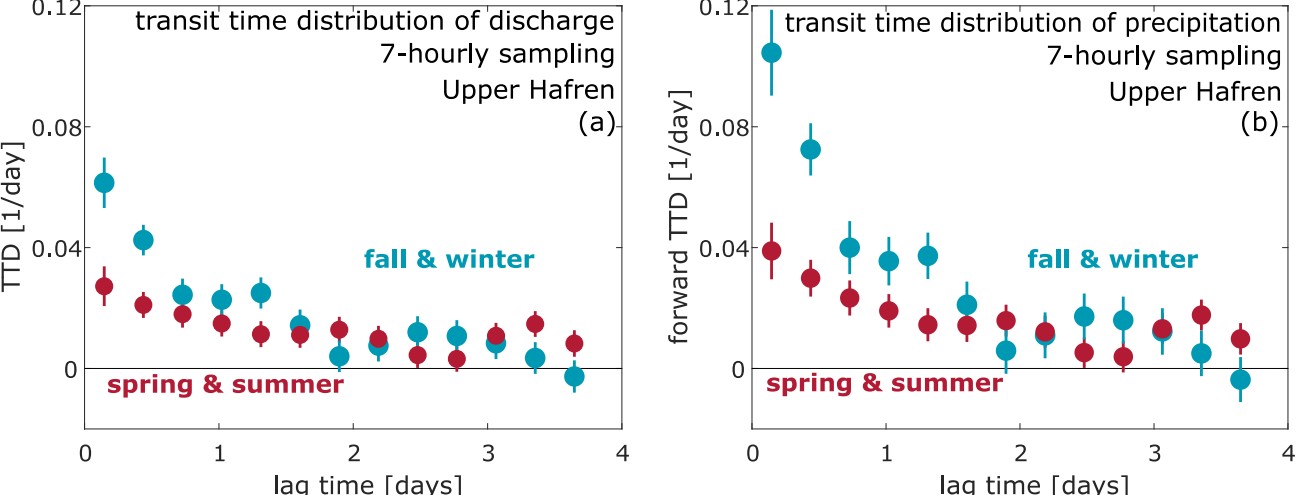

**Figure 12: Transit time distributions of discharge («backward» TTDs, a), and precipitation («forward» TTDs, b) calculated from 7-hourly oxygen-18 measurements at Upper Hafren for the months of September–February (fall & winter, blue), and March–August (spring & summer, red). Fall/winter TTDs exhibit stronger coupling between precipitation and streamflow than spring/summer TTDs do, over lag times up to 1-1.5 days but not longer. Fall/winter TTDs have larger error bars because there are more gaps in the source data.**





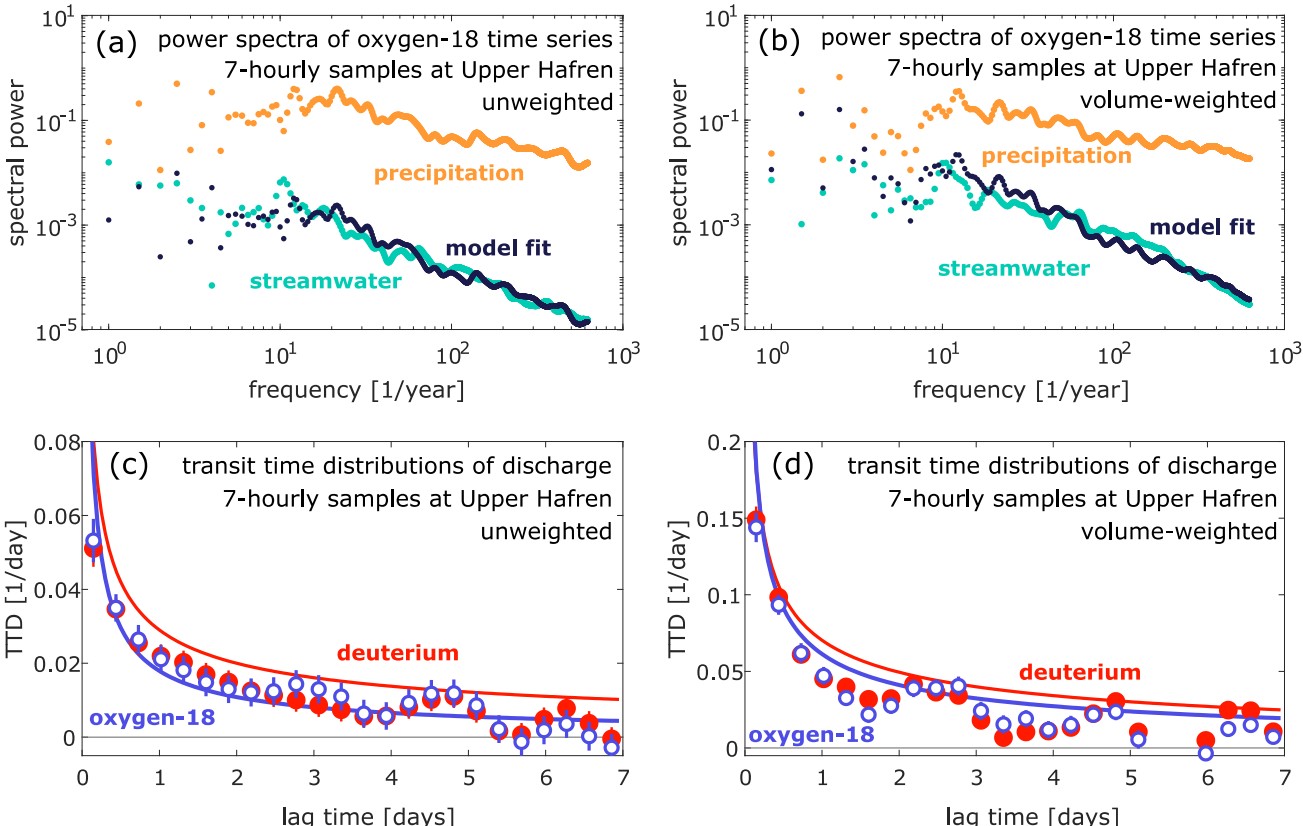

**Figure 13.** Fits of convolution models to unweighted (a) and volume-weighted (b) power spectra of oxygen-18 in precipitation and streamwater, and gamma distributions estimated by spectral fitting for both deuterium (red lines) and oxygen-18 (blue lines), compared to ensemble hydrograph separation estimates (dots, with standard errors) of unweighted (c) and volume-weighted (d) transit time distributions.