# Peer review of "New water fractions and transit time distributions at Plynlimon, Wales, estimated from stable water isotopes in precipitation and streamflow"

_Hydrology and Earth System Sciences, 2019_

## Referee Comment (RC1) · Markus Hrachowitz (Referee) · 17 Jun 2019

In this manuscript, the authors present high-temporal-resolution data sets of stable water isotope compositions in precipitation and streamflow for the Plynlimon research catchment. They then use these data to demonstrate its value for the characterization of catchment-scale transport characteristics in the form of "new water fractions" and transit time distributions. The paper is well-written and offers a detailed description and analysis of the presented data. In particular the comparison of the new 7-hourly data with previously collected weekly data gives the reader rare and interesting insights

into value of high resolution sampling. I would thus be more than glad to see this paper eventually published. However, I do have a few comments and questions, which I hope will help the authors to further strengthen the manuscript.

(1) I was a bit surprised by the discussion of the differences between "new water fractions" from 7-hourly and weekly samples, respectively (in particular, sections 5.1 and 5.3, together with figures 8-10). The way the analysis is presented now, it seems to the reader that it should be a surprise that the "new water fraction" increases with increased sampling interval. Of course, this is purely related to an ambiguous definition of "new water": the longer the time interval considered as "new", the more water label as "new" will reach the stream. Therefore, phrases such as "Which new water fractions are the correct ones [. . .]" (p.13,l.16) are very surprising. Instead, the reader may benefit more from this analysis and the concept of "new water", if this inherent ambiguity was clearly stated and explained upfront and the effects of it then shown in the subsequent analysis. It may thus be more informative to first provide an unambiguous definition (e.g. new water = 7 (or 14)-days sampling) and to then show a figure in section 5.3 with a direct comparison of the 7-day(!) or 14-day water fraction - as inferred from both, aggregated 7-hr sampling intervals and the weekly intervals, respectively. This would directly illustrate the gain of information when switching from low- to high-resolution sampling. Ideally, they would be identical. But are they?

(2) Related to the above, the discussion and treatment of what the authors refer to as "dry deposition" of chloride could benefit from a bit more detail. If I understood correctly, samples with high chloride concentrations are removed from the analysis. This can of course be done. However, I think it would be important to remind the reader that this is only a meaningful thing to do as long as the "new water fractions" and/or transit time distributions sought are limited to very short time periods. The longer the definition of "new water" or the transit times of interest (here: up to 7 days; Fig.13), the more uncertainties the exclusion of these concentrations will introduce into the analysis. Why? Even if entering the catchment by dry deposition, the chloride

mass deposited will not disappear and will be transported through the system with the subsequent rainfall events to eventually reach the stream. I may have missed something, but should dry deposition not, at least to some degree, be accounted for when considering volume-weighted estimates?

(3) It is great to see that the authors also provide an analysis of transit time distributions and their sensitivity to changes in wetness conditions and season. However, the sections 5.4-5.6 could strongly benefit from a bit more context. This sort of analysis has been done earlier, albeit with different methods, both in Plynlimon (e.g. Benettin et al., 2015; Harman, 2015) and elsewhere (e.g. Heidbuechel et al., 2012; Hrachowitz et al., 2013; van der Velde et al., 2015; von Freyberg et al., 2018). It may be interesting to compare the results and interpretations of this manuscript to the findings of at least these previous papers.

. Minor points:

p.2,l.8-9: "Because these tracers do not react...". We do not have any really passive tracers. The tracers we use are essentially all subject to some non-passive behaviour (as the authors also acknowledge somewhere later in the manuscript). Please rephrase.

p.2,l.28: I agree, but it may be interesting for the reader to add an explanation of why this may be beneficial.

p.2,l.32: Agreed. But I thought Kirchner et al. (2010) did not only ask the question but also provided some interesting insights. Please rephrase.

p.3,l.2: "...if the evaporated waters then evaporate completely...". Not sure I understand what you want to express here.

p.3,l.4: agreed, but this is only one possible effect on isotopes. Maybe rephrase to make this clearer. In addition, was it necessary to correct for altitude here? If yes, how was it done?

p.3,l.10-11: or where anthropogenic chloride inputs can be estimated (e.g. fertilizer; Hrachowitz et al., 2015)

p.3,l.18: if they are both transported conservatively(!) with the water then they *need to* yield similar results.

p.3,l.25: please provide references, e.g. Neal et al (2013) or Kirchner and Neal (2013) would fit nicely in here.

p.7,l.14-17: If 65% of the samples were subject to overflow and if the intra-interval isotope variations can be considerable, how reliable is the subsequent analysis then? This would warrant some discussion later on in the manuscript.

p.8,l.10: Kirchner et al. (2004) would fit nicely as reference here.

p.10,l.1: "can" or "are"?

p.12,l.10: ". . .less than 3% of streamflow. . .". When? On average? Or during a specific period?

p.12,l.13-15,21-22: this is obvious. See comment (1) – perhaps a better idea to make this the starting point and then illustrate the effects of it.

p.13,l.18-20: agreed. But should this not be a standard procedure at least since Niemi (1977)?

p.14,l.23-24: see also Hrachowitz et al. (2015)

. References:

Benettin, P., Kirchner, J. W., Rinaldo, A., & Botter, G. (2015). Modeling chloride transport using travel time distributions at Plynlimon, Wales. Water Resources Research, 51(5), 3259-3276.

Harman, C. J. (2015). Time-variable transit time distributions and transport: Theory and application to storage-dependent transport of chloride in a watershed. Water

Resources Research, 51(1), 1-30.

Heidbüchel, I., Troch, P. A., Lyon, S. W., & Weiler, M. (2012). The master transit time distribution of variable flow systems. Water Resources Research, 48(6).

Hrachowitz, M., Savenije, H., Bogaard, T. A., Tetzlaff, D., & Soulsby, C. (2013). What can flux tracking teach us about water age distribution patterns and their temporal dynamics?. Hydrology and Earth System Sciences 17, 533-564.

Hrachowitz, M., Fovet, O., Ruiz, L., & Savenije, H. H. (2015). Transit time distributions, legacy contamination and variability in biogeochemical $1/f\alpha$ scaling: how are hydrological response dynamics linked to water quality at the catchment scale?. Hydrological Processes, 29(25), 5241-5256.

Kirchner, J. W., & Neal, C. (2013). Universal fractal scaling in stream chemistry and its implications for solute transport and water quality trend detection. Proceedings of the National Academy of Sciences, 110(30), 12213-12218.

Kirchner, J. W., Feng, X., Neal, C., & Robson, A. J. (2004). The fine structure of water‐quality dynamics: The (high‐frequency) wave of the future. Hydrological Processes, 18(7), 1353-1359.

Kirchner, J. W., Tetzlaff, D., & Soulsby, C. (2010). Comparing chloride and water isotopes as hydrological tracers in two Scottish catchments. Hydrological Processes, 24(12), 1631-1645.

Neal, C., Reynolds, B., Kirchner, J. W., Rowland, P., Norris, D., Sleep, D., ... & Vincent, C. (2013). High‐frequency precipitation and stream water quality time series from Plynlimon, Wales: an openly accessible data resource spanning the periodic table. Hydrological Processes, 27(17), 2531-2539.

Niemi, A. J. (1977). Residence time distributions of variable flow processes. The International Journal of Applied Radiation and Isotopes, 28(10-11), 855-860.

van der Velde, Y., Heidbüchel, I., Lyon, S. W., Nyberg, L., Rodhe, A., Bishop, K., & Troch, P. A. (2015). Consequences of mixing assumptions for time‐variable travel time distributions. Hydrological Processes, 29(16), 3460-3474.

von Freyberg, J., Allen, S. T., Seeger, S., Weiler, M., & Kirchner, J. W. (2018). Sensitivity of young water fractions to hydro-climatic forcing and landscape properties across 22 Swiss catchments. Hydrol. Earth Syst. Sci, 22, 3841-3861.

---

## Author Comment (AC1) · 9 Jul 2019

**Response to the interactive comment of M. Hrachowitz on**

"New water fractions and transit time distributions at Plynlimon, Wales, estimated from stable water isotopes in precipitation and streamflow" by Julia L. A. Knapp et al.

We would like to thank Dr. Hrachowitz for reviewing our manuscript, and for his helpful comments. His comment on the definition of the "new water fraction" reveal the need for further explanations on our part. In the revised version, we will strengthen the manuscript by including a clearer explanation of the concept of "new water" and the differences between the new water fractions obtained from 7-hourly and weekly data. Please find our responses to the comments below. The comments provided by the reviewer are shown in italics, and our responses in regular font. Changes we will make in the revised manuscript are underlined.

*In this manuscript, the authors present high-temporal-resolution data sets of stable water isotope compositions in precipitation and streamflow for the Plynlimon research catchment. They then use these data to demonstrate its value for the characterization of catchment-scale transport characteristics in the form of "new water fractions" and transit time distributions. The paper is well-written and offers a detailed description and analysis of the presented data. In particular the comparison of the new 7-hourly data with previously collected weekly data gives the reader rare and interesting insights into value of high resolution sampling. I would thus be more than glad to see this paper eventually published. However, I do have a few comments and questions, which I hope will help the authors to further strengthen the manuscript.*

We thank Dr. Hrachowitz for the positive assessment of our work and his thoughtful comments.

**General Comments**

*(1) I was a bit surprised by the discussion of the differences between "new water fractions" from 7-hourly and weekly samples, respectively (in particular, sections 5.1 and 5.3, together with figures 8-10). The way the analysis is presented now, it seems to the reader that it should be a surprise that the "new water fraction" increases with in-creased sampling interval. Of course, this is purely related to an ambiguous definition of "new water": the longer the time interval considered as "new", the more water label as "new" will reach the stream. Therefore, phrases such as "Which new water fractions are the correct ones [...]" (p.13,l.16) are very surprising. Instead, the reader may benefit more from this analysis and the concept of "new water", if this inherent ambiguity was clearly stated and explained upfront and the effects of it then shown in the subsequent analysis. It may thus be more informative to first provide an unambiguous definition(e.g. new water = 7 (or 14)-days sampling) and to then show a figure in section 5.3with a direct comparison of the 7-day(!) or 14-day water fraction - as inferred from both, aggregated 7-hr sampling intervals and the weekly intervals, respectively. This would directly illustrate the gain of information when switching from low- to high-resolution sampling. Ideally, they would be identical. But are they?*

Dr. Hrachowitz is correct that the increase of new water fractions with sampling frequency is not surprising, because the magnitude of the new water fraction is an

inherent function of the sampling frequency. For this very reason, however, we do not believe that an unambiguous definition of new water (as, e.g., weekly new water) would be beneficial. Instead, the whole concept of "new water", through its way of estimation, is not locked to a specific time scale. But the result of a new water fraction calculation will inherently depend on the tracer sampling interval. The simplest way to make this clear is to embed the time scale in the label that is used for a new water fraction. Thus the "weekly new water fraction" as obtained from weekly sampling, and the "daily new water fraction", which would be obtained from daily sampling, are both examples of new water fractions, but there is no single "THE new water fraction" that is independent of the measurement time base.

We do agree with Dr. Hrachowitz that we should be more upfront with this, rather than presenting it only in the results. We will therefore include a paragraph in the "Calculation methods" section 4.1 of new water fractions, explaining this: "New water fractions assess this correlation on the time scale of the sampling frequency and are thus intrinsically tied to it. New water fractions calculated from weekly sampling are "weekly new water fractions", and express the ensemble average contribution to streamflow from precipitation that fell in the previous week. New water fractions calculated from 7-hourly sampling, or "7-hourly new water fractions", will be inherently smaller because they express the contribution to streamflow from precipitation that fell in the previous 7 hours instead of the previous week. As these examples show, new water fractions calculated for time series of different sampling frequencies will differ in both their magnitude and meaning, with smaller new water fractions obtained from higher-frequency sampling. The longer the sampling interval, the more precipitation labeled as "new" will have reached the stream by the time of sampling." We will furthermore clarify that new water fractions obtained from different sampling intervals mean something different in the updated version of the manuscript by using "weekly new water fraction" and "7-hourly new water fraction" throughout the manuscript, and only use the general term "new water fraction" when no specific time scale is considered.

The phrase "Which new water fractions are the correct ones…" (p.13 l.16) was focused on the differences between event new water fraction, new water fraction for all time steps, and the new water fraction of precipitation, NOT on the difference in new water fractions obtained from different sampling intervals. We will rephrase the sentences to make this clearer: "Whether event new water fractions, new water fractions for all time steps, or new water fractions of precipitation should be calculated…"

*(2) Related to the above, the discussion and treatment of what the authors refer to as "dry deposition" of chloride could benefit from a bit more detail. If I understood correctly, samples with high chloride concentrations are removed from the analysis. This can of course be done. However, I think it would be important to remind the reader that this is only a meaningful thing to do as long as the "new water fractions" and/or transit time distributions sought are limited to very short time periods. The longer the definition of "new water" or the transit times of interest (here: up to 7 days;Fig.13), the more uncertainties the exclusion of these concentrations will introduce into the analysis. Why? Even if entering the catchment by dry deposition, the chloride mass deposited will not disappear and*

*will be transported through the system with the subsequent rainfall events to eventually reach the stream. I may have missed something, but should dry deposition not, at least to some degree, be accounted for when considering volume-weighted estimates?*

To remove the effect of dry deposition of chloride, we excluded samples from the analysis of the 7-hourly data with very high chloride concentrations in very small-volume samples (details on this approach are provided in the supplemental information). We took this approach, because our analysis using ensemble hydrograph separation is based on the assessment of the correlation between the input and output concentrations, rather than a mass balance. Consequently, the timing of the input is of greater relevance than the total mass. Due to the large funnel size, a few rain drops are sufficient to create a precipitation sample with enough volume to be analyzable, but these few rain drops are probably not enough to wash all of the deposited chloride into the catchment (please note, large sample volumes, indicating larger rain events, were not removed during the dry deposition correction). If we use the data as is, however, the actual input to the catchment will occur later (i.e., during the next rain event) than when the dry deposition is captured in the samples (i.e., with the first few rain drops following the dry deposition), leading to a mismatch in timing between real-world processes and data. Furthermore, including the dry deposition affected samples would result in a large effect of a handful of samples with very low volumes but extremely high concentration in the analysis through ensemble hydrograph separation.

As Dr. Hrachowitz correctly points out, the approach used to filter out dry deposition effects does not account for the mass of chloride entering the catchment through dry deposition. To conserve this mass, and get the timing of the actual input to the catchment right, it could have been a valid approach to identify dry-deposition affected samples and move the dry-deposited mass of chloride to the next observed rain event in the data set. Since the actual masses are small, however, this would likely not have affected our analysis by much. Furthermore, since we have no way of being certain than a sample is actually affected by dry deposition, this may have biased the analysis more than the exclusion of samples potentially affected by dry deposition.

Samples strongly affected by dry deposition usually contain very small sample volumes, with very high concentrations. In the ensemble hydrograph separation approach, volume-weighting is achieved through discharge-weighting, rather than weighting by precipitation volumes. Low precipitation volumes are often, but not always, associated with low discharge values. Consequently, these dry deposition-affected samples may still get a substantial weight even in volume-weighted new water fraction calculations. A better approach to remove these low-volume dry deposition samples is through the precipitation threshold, below which samples are excluded from analysis. As Fig. 8 shows, however, this was not enough to remove all dry deposition affected samples in case of chloride.

We will add a more detailed explanation in Sect. 5.3: "The analysis thus showed that chloride may be a suitable passive tracer, if potential effects of dry deposition are removed. However, it is important to note that the filtering approach for dry deposition employed here was not empirically validated and was not based on physical effects like wind speed or direction. Furthermore, the removal of dry-deposition affected

samples leads to a reduced mass recovery. In the ensemble hydrograph separation approach, this has only a small effect, as only the correlation between the input and output signal is assessed. In other approaches, however, a correct mass balance is essential. Therefore, we argue that the stable water isotope data provide a better and more reliable data set to quantify catchment characteristics, mixing and storage processes in the catchment."

*(3) It is great to see that the authors also provide an analysis of transit time distributions and their sensitivity to changes in wetness conditions and season. However, the sections 5.4-5.6 could strongly benefit from a bit more context. This sort of analysis has been done earlier, albeit with different methods, both in Plynlimon (e.g. Benettin et al., 2015; Harman, 2015) and elsewhere (e.g. Heidbuechel et al., 2012; Hrachowitz et al., 2013; van der Velde et al., 2015; von Freyberg et al., 2018). It may be interesting to compare the results and interpretations of this manuscript to the findings of at least these previous papers.*

We will include an additional comparison to some of the mentioned papers in the revised manuscript at the end of Sect. 5.6: "Transit time distributions have previously been assessed at Plynlimon from chloride data using StorAge Selection functions. Benettin et al. (2015) calibrated a two-box model to the data and obtained a mean transit time at the Upper Hafren catchment of approximately 1.5 yrs. Conversely, Harman (2014) used rank StorAge Selection functions at the Lower Hafren, an approach which requires making assumptions about the parametric shape of the transit time distribution. If a gamma distribution was assumed, Harman (2014) found median transit times of 400 and 550 days for fixed and storage-dependent calculations, respectively. Our approach, on the other hand, depends more directly on data. In spite of these substantially different analyses, we obtained mean transit times that are relatively similar to those found by Benettin et al. (2015) and Harman (2014).
Our approach also resulted in similar shapes of the transit time distributions. Benettin et al. (2015) found that the marginal transit time distribution closely resembled a gamma distribution with the shape factor of $k = 0.5$, while Harman (2014) obtained a shape factor of $k = 0.52$ when enforcing a gamma distribution. This indicates the general plausibility of the underlying shape function, even though the shape factors $k$ obtained from fitting to volume-weighted power spectra in our study varied between 0.40-0.54. These similarities are noteworthy because our approach estimates the short-time tail of the transit time distribution directly from tracer data; the shape of the distribution is not specified in advance."

**Minor points.**
*p.2,l.8-9: "Because these tracers do not react...". We do not have any really passive tracers. The tracers we use are essentially all subject to some non-passive behaviour (as the authors also acknowledge somewhere later in the manuscript). Please rephrase.*

We agree. We will modify this statement in the revised manuscript: "Because the interaction of these tracers with their environment is limited,…"

*p.2,l.28: I agree, but it may be interesting for the reader to add an explanation of why this may be beneficial.*

We will add "…, because the variations of the water fluxes as drivers of the underlying processes need to be reflected in the sampling" here.

*p.2,l.32: Agreed. But I thought Kirchner et al. (2010) did not only ask the question but also provided some interesting insights. Please rephrase.*

True. We will (a) move the Kirchner et al. (2010) citation from here to line 19 on the same page, (b) add more details on the findings of Kirchner et al. (2010) in line 19: "In this context, Kirchner et al. (2010) found that power spectra for both tracers exhibited similar patterns of damping of fluctuations from precipitation to streamwater, but the damping was stronger for oxygen-18 than chloride."

*p.3,l.2: "...if the evaporated waters then evaporate completely...". Not sure I under-stand what you want to express here.*

We rephrase: "If the soil water and precipitation affected by evaporation is subsequently evaporated completely, …."

*p.3,l.4: agreed, but this is only one possible effect on isotopes. Maybe rephrase to make this clearer. In addition, was it necessary to correct for altitude here? If yes, how was it done?*

We did not deem a correction for altitude effects necessary for two reasons: First of all, the location of the precipitation sampling is relatively representative of the catchment average elevation, particularly in case of the (Lower) Hafren catchment. Second, the overall relief in this landscape is relatively small (only 198 m in the Tanllwyth catchment, 188 m in the Upper Hafren catchment, and 382 m in the Hafren catchment). Furthermore, the correction for altitude effects is commonly done by adding a constant offset to the time series, based on the distribution of rainfall amounts at different altitudes across the catchment. Our analysis used in this study is based on the correlation of input and output signals, and therefore is not sensitive to constant offsets.

The dataset documentation currently provides information on the coordinates and altitudes of the sampling stations. To make it easier for other users of the data set to correct for altitude effects if necessary, we will add a sentence to the dataset documentation explaining where information on elevations and catchment boundaries can be found: "Further details on spatial extents and spot heights, as well as a digital terrain model are available from the Center of Ecology & Hydrology: https://catalogue.ceh.ac.uk/documents/91961a0f-3158-4d00-984d-91eb1e03e8bd."

*p.3,l.10-11: or where anthropogenic chloride inputs can be estimated (e.g. fertilizer; Hrachowitz et al., 2015)*

Yes, but only if the fertilizer input is homogeneously distributed in space. This is rarely the case, for which reason we prefer not to go into details in the manuscript.

*p.3,l.18: if they are both transported conservatively(!) with the water then they \*need to\* yield similar results.*

No, not necessarily. If processes before or after the transport through the catchment substantially affect the tracer signals, e.g. evaporation effects, or through dry deposition or evapoconcentration, the tracers would yield different results.

*p.3,l.25: please provide references, e.g. Neal et al (2013) or Kirchner and Neal (2013) would fit nicely in here.*

Agreed. We will add references to Neal et al. (2013a), Neal et al. (2013b) and Norris et al. (2017).

*p.7,l.14-17: If 65% of the samples were subject to overflow and if the intra-interval isotope variations can be considerable, how reliable is the subsequent analysis then? This would warrant some discussion later on in the manuscript.*

We agree that this is an important point. However, the overflow does not pose a significant problem, if the captured sample nevertheless represents the composition of the rainfall event as function of precipitation amount. To verify this, we compared volume-weighted averages of the 7-hourly samples to the measured weekly bulk samples (which did not have the same overflow problems) covering the same time intervals. These agree quite well (see Fig. 1 below).
We will modify the relevant section in the revised manuscript as follows: "To verify that this did not substantially affect the data, we compared volume-weighted averages of the 7-hourly data to the corresponding weekly bulk precipitation samples and found good agreement. This suggests that isotopic mass balances derived from these data are reliable, even though samples that overflowed comprise the great majority of the total rainfall, and within-event variations in precipitation isotopes can be large (Munksgaard et al., 2012; von Freyberg et al., 2017)."

[Figure]

Figure 1: Comparison of weekly bulk samples and volume-weighted weekly averages from 7-hourly samples for deuterium (top left), oxygen-18 (top right), chloride (bottom left) and the precipitation volume (bottom right). The black line indicates the 1:1 line.

*p.8,l.10: Kirchner et al. (2004) would fit nicely as reference here.*

We agree. We will add the reference to the revised manuscript.

*p.10,l.1: "can" or "are"?*

"can" is correct, as we provide volume-weighted and unweighted estimates of new water fractions in the manuscript (also see below).

*p.12,l.10: "...less than 3% of streamflow...". When? On average? Or during a specific period?*

On average. We will modify the sentence to make this clearer: "7-hourly new water fractions (calculated from 7-hourly isotope data) show that on average, slightly less than 3% of streamflow was made up of precipitation that fell within the last 7 hours. Weekly new water fractions (calculated from weekly isotope data) show that on

average 13-15% of streamflow consisted of precipitation that fell within the last week. (For both sampling frequencies, these are volume-weighted new water fractions for all time steps, QF*new, and thus include periods where no precipitation fell).”

*p.12,l.13-15,21-22: this is obvious. See comment (1) – perhaps a better idea to make this the starting point and then illustrate the effects of it.*

Yes, we agree. As discussed above, we will add a short paragraph to the calculation methods of new water fractions, and rephrase the terminology throughout the paper. Regarding volume-weighting (lines 21-22), we prefer to keep the explanation here, as the results nicely underline our statement.

*p.13,l.18-20: agreed. But should this not be a standard procedure at least since Niemi(1977)?*

We believe that both volume-weighted and unweighted new water fractions can provide interesting insights into catchment processes. Whereas volume-weighted new water fractions will be sensitive to the few times with very high flows, and thus provide information mainly about these time points, unweighted new water fractions provide information on average behavior over all time points, not just when the catchment is very wet.

p.14,l.23-24: see also Hrachowitz et al. (2015)

We agree. We will add the following sentence to the revised version of the manuscript: “The sensitivity of chloride to evapoconcentration and its substantial effect of the damping of chloride signals were also shown by Hrachowitz et al. (2015).”

**Referenced Literature**

Benettin, P., Kirchner, J. W., Rinaldo, A., & Botter, G.: Modeling chloride transport using travel time distributions at Plynlimon, Wales. *Water Resources Research*, *51*(5), 3259-3276, 2015

Harman, C. J.: Time-variable transit time distributions and transport: Theory and application to storage-dependent transport of chloride in a watershed. *Water Resources Research*, *51*(1), 1-30, 2015.

Hrachowitz, M., Fovet, O., Ruiz, L., & Savenije, H. H.: Transit time distributions, legacy contamination and variability in biogeochemical 1/fα scaling: how are hydrological response dynamics linked to water quality at the catchment scale? *Hydrological Processes*, *29*(25), 5241-5256, 2015.

Kirchner, J. W., Feng, X., Neal, C., & Robson, A. J.: The fine structure of water-quality dynamics: The (high-frequency) wave of the future. *Hydrological Processes*, *18*(7), 1353-1359, 2004.

Kirchner, J. W., Tetzlaff, D., & Soulsby, C.: Comparing chloride and water isotopes as hydrological tracers in two Scottish catchments. *Hydrological Processes*, *24*(12), 1631-1645, 2010.

Munksgaard, N., Wurster, C., Bass, A., and Bird, M.: Extreme short-term stable isotope variability revealed by continuous rainwater analysis, Hydrol. Process., 26, 3630-3634, 2012.

Neal, C., Kirchner, J., and Reynolds, B.: Plynlimon research catchment high-frequency hydrochemistry data, NERC Environmental Information Data Centre, 2013a.

Neal, C., Kirchner, J., and Reynolds, B.: Plynlimon research catchment hydrochemistry, NERC Environmental Information Data Centre, 2013b.

Norris, D. A., Harvey, R., Winterbourn, J. M., Hughes, S., Lebron, I., Thacker, S. A., Lawlor, A. J., Carter, H. T., Patel, M., Keenan, P. O., Pereira, M. G., Cosby, B. J., Reynolds, B., Grant, S. J., Pomeroy, I., Hinton, C., Spinney, K., Peters, T. D., and Callahan, B.: Plynlimon research catchment hydrochemistry (2011-2016), NERC Environmental Information Data Centre, 2017.

von Freyberg, J., Studer, B., and Kirchner, J. W.: A lab in the field: high-frequency analysis of water quality and stable isotopes in stream water and precipitation, Hydrol. Earth Syst. Sci., 21, 1721-1739, 2017.

---

## Referee Comment (RC2) · Nigel Roulet (Referee) · 17 Aug 2019

This a well-written and well-argued paper. It will be a valuable contribution the runoff literature, particularly the interpretation of separating storm flow components. The authors use a high temporal resolution isotope and chloride data set for several catchments in Plynlimon, Wales to address a number of questions related to the separation of new (event) water from older 'stored' water in runoff. They calculate transit times, fraction of event water, and spectral filtering to attempt to tease out catchment transport and storage processes. The paper uses inference from the outflow record and has no

physical information to actually figure out transport and storage.

The results are not overly surprising – one needs to define well what one is analyzing and the appropriateness of various define characteristics are assessed relative to the research questions being asked. This seems obvious. The sensitivity of the results to the frequency of sampling is also not surprising but this is a nice empirical analysis of the effect. This study is a good example of the importance of stored water to storm runoff. It is also show a reassuring similarity between isotopic tracers – the isotopes producing essential the same result but Cl yields less event water than the isotopes. I suspect this is because the rain water signal for Cl is derived rather than directly measured as an input signal. This is not the case with the isotopes.

The paper is timely. As the authors state the high resolution data set they use is unique but with new, reasonably priced, technologies for measuring isotopes in a semi-continuous manner coming on line, the issues this paper raises will be very important.

Pg 3 – ln 1 "gold standard". There is no such thing in hydrology for this kind of word. One would have to understand the flow system to get one. Even in the constructed settings the variability is a problem. Not sure this term is useful – will it ever be obtained?

Pg 3 – Ln 20 – 25. This statement is correct but could be a little more forwarding looking to the future.

Pg 11 ln 1 hints at this future. Why not be explicit?

Pg 11 – ln 19-21. Not sure I understand why you did not filter the Cl? Something is not making sense to me here.

Pg 13 – ln 22-26 The filtering issue again. How good it is depends on how you can eliminate the dry deposition issue. Can you elaborate?

Pg 14 Ln 8-15 Same issue. Not sure why the dry deposition would make the new fraction smaller?

Pg. 14 ln 26-34. This seems obvious and suggests that operational definitions need to be specified so in the future we know what we are comparing. Why not be more explicit in the definition of thresholds.

Nigel Roulet, McGill University, August 2019

---

## Author Comment (AC2) · 19 Aug 2019

**Response to the interactive comment of N. Roulet on**
"New water fractions and transit time distributions at Plynlimon, Wales, estimated from stable water isotopes in precipitation and streamflow" by Julia L. A. Knapp et al.

We want to thank Nigel Roulet for reviewing our manuscript and for providing helpful comment. Please find our responses to the comments below. The comments provided by Prof. Roulet are shown in italics, and our responses in regular font. Changes we will make in the revised manuscript are underlined.

*This a well-written and well-argued paper. It will be a valuable contribution the runoff literature, particularly the interpretation of separating storm flow components. The authors use a high temporal resolution isotope and chloride data set for several catchments in Plynlimon, Wales to address a number of questions related to the separation of new (event) water from older 'stored' water in runoff. They calculate transit times, fraction of event water, and spectral filtering to attempt to tease out catchment transport and storage processes. The paper uses inference from the outflow record and has no physical information to actually figure out transport and storage.*

*The results are not overly surprising – one needs to define well what one is analyzing and the appropriateness of various define characteristics are assessed relative to the research questions being asked. This seems obvious. The sensitivity of the results to the frequency of sampling is also not surprising but this is a nice empirical analysis of the effect. This study is a good example of the importance of stored water to storm runoff. It is also show a reassuring similarity between isotopic tracers – the isotopes producing essential the same result but Cl yields less event water than the isotopes. I suspect this is because the rain water signal for Cl is derived rather than directly measured as an input signal. This is not the case with the isotopes.*

*The paper is timely. As the authors state the high resolution data set they use is unique but with new, reasonably priced, technologies for measuring isotopes in a semi-continuous manner coming on line, the issues this paper raises will be very important.*

We thank Prof. Roulet for the positive assessment of the manuscript.

*Pg 3 – ln 1 "gold standard". There is no such thing in hydrology for this kind of word. One would have to understand the flow system to get one. Even in the constructed settings the variability is a problem. Not sure this term is useful – will it ever be obtained?*

We agree that (unfortunately) no "gold standard" tracer exists in hydrology (and our point was that neither isotopes nor chloride fills this role). We will change the sentence to "..., as both tracers suffer from different shortcomings" in the revised version of the manuscript.

*Pg 3 – Ln 20 – 25. This statement is correct but could be a little more forwarding looking to the future.*

We assume that "more forward looking to the future" refers to the more widespread availability of isotope measurements, including new technology enabling the semi-continuous measurements of stable water isotopes in a more automated manner. This in an important point, and we will include this in the revised version of the manuscript, albeit not at this point in the introduction, but in the conclusion (p.19, l.3, following "… we conclude that stable water isotopes provide a more reliable basis for quantifying catchment transport timescales."): "…, especially in the light of novel technology, enabling semi-continuous measurements of stable water isotopes in an automated manner (e.g., von Freyberg et al., 2017)."

*Pg 11 ln 1 hints at this future. Why not be explicit?*

We agree that the advancements with regard to stable water isotope technology should be mentioned explicitly. This section of the manuscript, however, refers to the analysis of chloride data. Instead, we will include a note toward the end of the paper. Please see our response to the previous comment for details.

*Pg 11 – ln 19-21. Not sure I understand why you did not filter the Cl? Something is not making sense to me here.*

We did not filter the weekly chloride samples, because we expected the effect of dry deposition to be less important in these samples. For one, the funnel was smaller, meaning more absolute amount of rain was needed to make up a measurable sample than for the 7-hourly sampling. Therefore, any dry deposition was much more diluted in the weekly sampling compared to the 7-hourly sampling. Due to the longer sampling interval, we also expected dry deposition to have some effect on nearly every sample. Therefore, it would also have been more difficult to identify samples with a substantial effect of dry deposition and exclude them based on empirical criteria. For these reasons, we decided to merely perform a general outlier removal on the weekly chloride data to remove samples with unrealistically high concentrations.

*Pg 13 – ln 22-26 The filtering issue again. How good it is depends on how you can eliminate the dry deposition issue. Can you elaborate?*

This is correct. Following suggestions from the first reviewer of this manuscript, we have decided to add additional detail on the dry deposition filtering in Sect. 5.3. We will expand this additional comment to include the point made by Prof. Roulet: "The analysis thus showed that chloride may be a suitable passive tracer, if potential effects of dry deposition are removed. The extent of the suitability of chloride as a tracer consequently depends on how well dry deposition effects can be identified and removed. However, it is important to note that the filtering approach for dry deposition

employed here was not empirically validated and was not based on physical effects like wind speed or direction. Furthermore, the removal of dry-deposition affected samples leads to a reduced mass recovery. In the ensemble hydrograph separation approach, this has only a small effect, as only the correlation between the input and output signal is assessed. In other approaches, however, a correct mass balance is essential. Therefore, we argue that the stable water isotope data provide a better and more reliable data set to quantify catchment characteristics as well as mixing and storage processes."

*Pg 14 Ln 8-15 Same issue. Not sure why the dry deposition would make the new fraction smaller?*

As explained in the manuscript, dry deposition leads to some few precipitation samples with unrealistically high concentrations. This has the effect of stretching the x-axis of the regression whose slope yields the new water fraction. As the x-axis becomes more and more extended by these high-concentration outliers, the regression slope becomes lower and lower, and consequently the calculated new water fraction is smaller.

*Pg. 14 ln 26-34. This seems obvious and suggests that operational definitions need to be specified so in the future we know what we are comparing. Why not be more explicit in the definition of thresholds.*

This paragraph discusses the precipitation threshold. As described in the manuscript, explicitly defining the threshold is not feasible, as it "depends on the frequency and intensity of rain events, as well as the sampling frequency". Strictly defining the magnitude of the threshold is therefore not a good idea. However, if different systems are compared, the precipitation thresholds should be comparable, e.g. lead to the exclusion of similar percentages of total precipitation. We will add this to the revised manuscript (at the end of line 29): "If different systems are compared, we recommend choosing a precipitation threshold that will exclude similar fractions of precipitation volumes and isotope samples. In our case, the precipitation threshold of 0.1 mm h$^{-1}$ led to an exclusion of…"

**Referenced Literature**

von Freyberg, J., Studer, B., and Kirchner, J. W.: A lab in the field: high-frequency analysis of water quality and stable isotopes in stream water and precipitation, Hydrol. Earth Syst. Sci., 21, 1721-1739, https://doi.org/10.5194/hess-21-1721-2017, 2017.